# Physiological synaptic activity and recognition memory require astroglial glutamine

Giselle Cheung [1,8], Danijela Bataveljic [1,9,12], Josien Visser[1,2,12], Naresh Kumar [3,12], Julien Moulard [1,2], Glenn Dallérac [1,10], Daria Mozheiko[1,2], Astrid Rollenhagen[4,5], Pascal Ezan[1], Cédric Mongin [3], Oana Chever [1,11], Alexis-Pierre Bemelmans [6], Joachim Lübke[4,5,7], Isabelle Leray[3] & Nathalie Rouach [1]✉

Presynaptic glutamate replenishment is fundamental to brain function. In high activity regimes, such as epileptic episodes, this process is thought to rely on the glutamate-glutamine cycle between neurons and astrocytes. However the presence of an astroglial glutamine supply, as well as its functional relevance in vivo in the healthy brain remain controversial, partly due to a lack of tools that can directly examine glutamine transfer. Here, we generated a fluorescent probe that tracks glutamine in live cells, which provides direct visual evidence of an activity-dependent glutamine supply from astroglial networks to pre-synaptic structures under physiological conditions. This mobilization is mediated by connexin43, an astroglial protein with both gap-junction and hemichannel functions, and is essential for synaptic transmission and object recognition memory. Our findings uncover an indispensable recruitment of astroglial glutamine in physiological synaptic activity and memory via an unconventional pathway, thus providing an astrocyte basis for cognitive processes.

[1] Neuroglial Interactions in Cerebral Physiology and Pathologies, Center for Interdisciplinary Research in Biology (CIRB), Collège de France, CNRS, INSERM, Labex Memolife, Université PSL, Paris, France. [2] Doctoral School N°158, Pierre and Marie Curie University, Paris, France. [3] Université Paris Saclay, ENS Paris-Saclay, CNRS, PPSM, 91190 Gif-sur-Yvette, France. [4] Institute for Neuroscience and Medicine INM-10, Research Center Jülich, Jülich, Germany. [5] Department of Psychiatry, Psychotherapy and Psychosomatics, Rheinisch-Westfaelische Technische Hochschule Aachen University, Aachen, Germany. [6] Commissariat à l'Energie Atomique et aux Energies Alternatives, Département de la Recherche Fondamentale, Institut de biologie François Jacob, Molecular Imaging Research Center and CNRS UMR9199, Université Paris-Sud, Neurodegenerative Diseases Laboratory, Fontenay-aux-Roses, France. [7] Jülich-Aachen Research Alliance Translational Brain Medicine, Aachen, Germany. [8] Present address: Institute of Science and Technology Austria, Klosterneuburg, Austria. [9] Present address: Institute for Physiology and Biochemistry Ivan Djaja, Faculty of Biology, University of Belgrade, Belgrade, Serbia. [10] Present address: Université Paris-Saclay, CNRS, Institut des Neurosciences Paris-Saclay, Gif-sur-Yvette, France. [11] Present address: Normandy University, UNIROUEN, INSERM, DC2N, 76000 Rouen, France. [12] These authors contributed equally: Danijela Bataveljic, Josien Visser, Naresh Kumar. ✉email: nathalie.rouach@college-de-france.fr

Neurotransmitter replenishment upon release of synaptic vesicles is essential for many neuronal functions. Although some neurotransmitters can be retrieved by neurons through various reuptake mechanisms, glutamate is mainly removed by perisynaptic astrocytes via glutamate transporters[1,2]. This removal of glutamate not only helps prevent excitotoxicity, but also acts as a local mechanism to recycle synaptic glutamate. The glutamate–glutamine cycle postulates that astrocytes take up glutamate from the synaptic cleft and transform it into glutamine, which is transported back to neurons to be converted to glutamate[3]. Such a mechanism would be important in vivo because de novo glutamine synthesis does not occur at nerve terminals[4,5]. In fact, it was estimated that, without replenishment, the pool of presynaptic glutamate would be exhausted within a minute of basal synaptic activity[6]. While this cycle has been implicated in epilepsy, a hyperactive state where the rate of glutamate release is greatly enhanced[7–9], its relevance to physiological conditions is still under debate due to conflicting reports. Early reports claimed that inhibition of glutamate to glutamine conversion or glutamine transport into neurons does not affect excitatory synaptic transmission or quantal size[10–12]. Chronic inhibition of glutamine synthetase was also found to have no effect on learning and memory in mice[13]. In contrast, other studies showed that presynaptic glutamine transport contributes to miniature EPSC amplitude[14] and astroglial glutamine synthesis sustains excitatory transmission[15]. A major step forward in helping to clarify the role of astrocytic glutamine would be to develop tools capable of directly assessing glutamine transfer. While radiolabeled-glutamine has previously been used for quantitative measurements of glutamine in brain samples[16,17], this approach offers poor spatial and temporal resolution. More recently, two genetically encoded FRET-sensors for glutamine have been developed for plant root tips[18] and cos7 cells[19]. Although useful in live preparations, these sensors depend on genetic manipulations, and cannot distinguish a directional flow of glutamine across adjacent cells. In the absence of tools that can track directional glutamine movement, a direct role for astroglial glutamine supply during physiological synaptic activity remains unclear. Here we develop a glutamine fluorescent probe, which allowed us to gather direct visual evidence for activity-dependent astrocyte-neuron glutamine transfer under physiological conditions. Further, we find that the role of glutamine is dependent on the opening of astroglial connexin 43 hemichannels upon activity, and provide evidence for the functional relevance of astroglial glutamine in physiological synaptic transmission and cognition.

## Results

### Generation of a fluorescent glutamine molecule.
We generated a fluorescent rhodamine-tagged glutamine molecule (RhGln) as a probe to directly visualize glutamine in live cells from brain tissues. RhGln was synthesized in a 5-step procedure (Fig. 1a and Supplementary Fig. 1), and is a small (0.64 kDa), brightly fluorescent, stable, water soluble, and non-hydrolysable glutamine analog. The fluorescent tag was conjugated to the amide side chain to prevent hydrolysis of glutamine to glutamate by glutaminase, as previously described for theanine[20], thereby enabling visualization of glutamine only but not its metabolites. The rhodamine moiety is flat and positively charged. According to the density functional theory (DFT) calculations, it is spaced at an optimal distance from glutamine leaving the glutamine function intact for its biological functions. Further characterization showed that RhGln has an absorption and emission maxima at 580 and 601 nm, respectively (Fig. 1b, c). Compared to the unconjugated rhodamine molecule (Rh101), RhGln showed a slight bathochromic shift of absorption and emission spectra and a fluorescence emission quantum yield of 0.6 (Fig. 1d, e).

### Activity-dependent mobilization of glutamine from astrocytes to presynaptic structures.
To visualize cellular transfer of astroglial glutamine, we dialyzed RhGln (0.8 mM, 20 min) into single CA1 astrocytes from hippocampal slices using the whole-cell patch-clamp technique, and observed diffusion of RhGln into neighboring cells (Fig. 2a), identified as GFP-positive astrocytes from Aldh1l1-eGFP mice (Fig. 2b). RhGln-labeling in neighboring astrocytes was located in soma and processes, including both main and fine processes, as revealed by RhGln dialysis of GFAP-eGFP astrocytes (Fig. 2c). RhGln-labeling was not due to extracellular uptake, as RhGln was delivered selectively and intracellularly to single astrocytes via a patch pipette and extracellular bulk loading of slices with RhGln did not label astrocytes (Supplementary Fig. 2). Using Aldh1l1-eGFP transgenic mice, in which the majority of astrocytes is labeled by GFP[21], we found that while CA1 pyramidal neurons take up RhGln, none of the 135 labeled astrocytes in the hippocampal CA1 region were loaded with RhGln (Supplementary Fig. 2a). In addition to this, no RhGln-labeling at the level of astroglial fine processes was found when loading was performed in acute slices from GFAP-eGFP mice, where these fine processes are visible by GFP-labeling (Supplementary Fig. 2b–d). The diffusion of RhGln into astroglial networks occurred via gap junction (GJ) channels, because it was inhibited by the GJ blocker carbenoxolone (CBX, Fig. 2a, e). We next investigated whether the transfer of astroglial glutamine was activity-dependent by comparing RhGln distribution during basal and evoked activity triggered by Schaffer collaterals repetitive stimulation (10 Hz, 30 s). The stimulation evoked field excitatory postsynaptic potentials (fEPSPs) and an astrocytic response (Fig. 2d). Surprisingly, we found that glutamine intercellular transfer was reduced in astrocytes upon stimulation (Fig. 2a, f), implying that synaptic activity induces a redistribution of glutamine away from astroglial networks. To test whether this redistribution is specific to RhGln, we dialyzed astrocytes with the rhodamine (Rh) dye alone (0.8 mM), and observed in basal conditions that Rh also diffused into neighboring cells to a similar extent as RhGln over 20 min (Fig. 2a, e). However, upon enhanced activity (10 Hz, 30 s), Rh diffusion was significantly enhanced, unlike the restriction observed using RhGln (Fig. 2a, f). This is reminiscent of previous observations of an activity-dependent trafficking of inert dyes in astroglial network[22–24]. Together, these data indicate that the activity-dependent redistribution of glutamine away from the astroglial network is specific for RhGln, as it is not observed with the unconjugated Rh dye.

We examined where glutamine was redistributed to at a subcellular level, and found that RhGln did not only spread into neighboring astrocytes, but was also present in punctate structures surrounding the patched astrocyte (Fig. 3a). Furthermore, the RhGln punctate labeling was much stronger and spread further upon stimulation (Fig. 3a, b, e), indicating an activity-dependent mobilization of glutamine away from the astrocyte soma. This activity-dependent punctate labeling was specific to RhGln as it was not observed with the Rh dye alone (Fig. 3a, c, e). It is also unlikely to result from activity-dependent morphological changes in astrocytes, since we found no change in soma size, domain area, and ramifications, as determined in GFP- and GFAP-positive astrocytes from Aldh1l1-eGFP mice (Supplementary Fig. 3). We also verified that CA1 synaptic transmission remained stable over the course of dye loading into a single astrocyte by monitoring fEPSP slope in response to repetitive synaptic stimulation (10 Hz) (Supplementary Fig. 4). This suggests that RhGln can be used to monitor the mobilization of glutamine without affecting neuronal glutamate replenishment from endogenous glutamine, important for synaptic activity. Because fine astroglial processes are mainly found near synaptic structures[25] and activity-dependent RhGln punctate labeling was

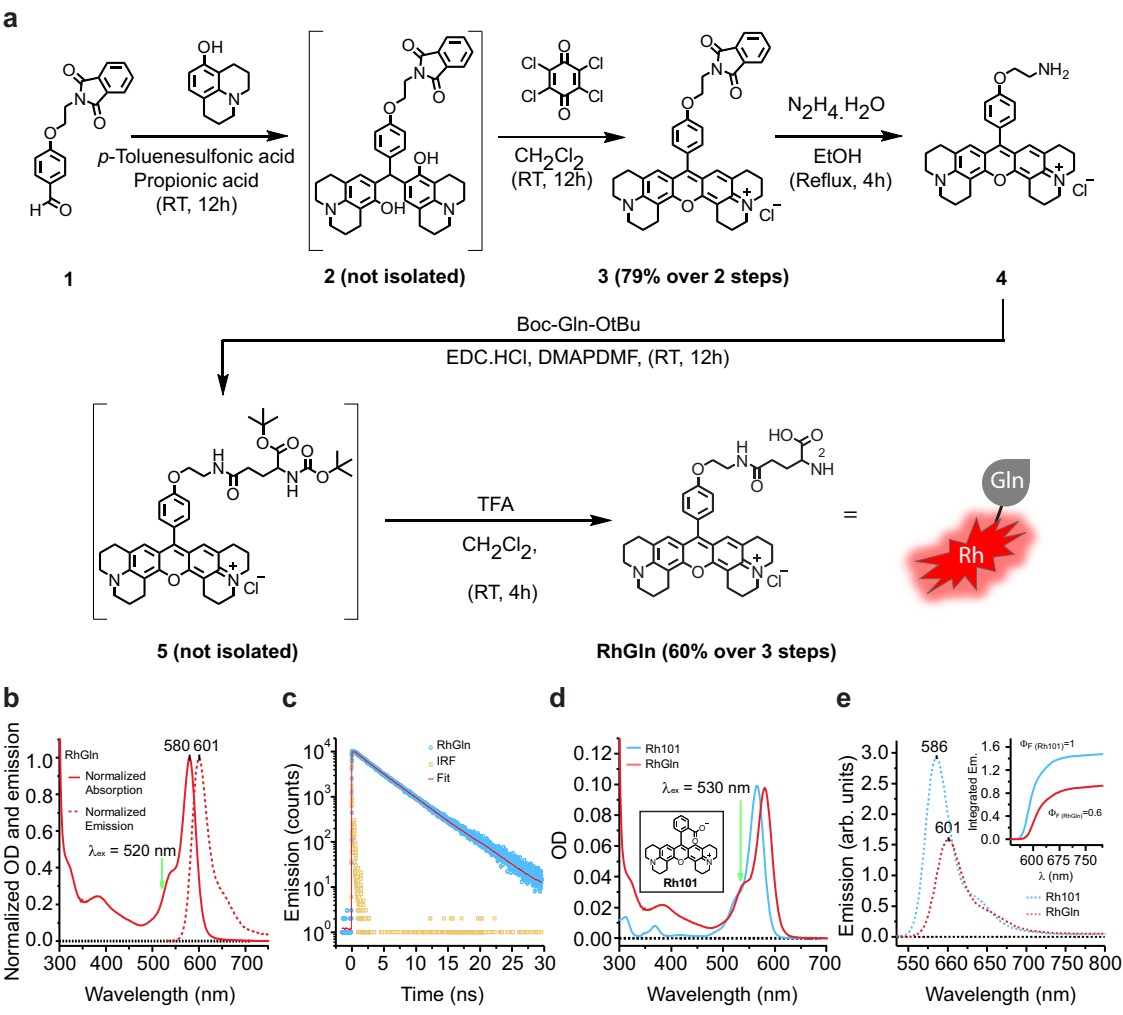

**Fig. 1 Synthesis and characterization of fluorescent rhodamine-tagged glutamine (RhGln) molecule. a** Fluorescent RhGln molecule was prepared using a 5-step chemical synthesis with 48% overall yield. **b** Characterization of steady-state electronic absorption (solid line) and emission (dotted line) spectra of RhGln at excitation wavelength ($\lambda_{ex}$) of 520 nm in intracellular solution at 20 °C, showing sharp absorption and emission maxima of 580 and 601 nm, respectively. **c** Fluorescence lifetime decay of RhGln in intracellular solution at $\lambda_{ex}$ of 520 nm (blue circles). The instrument response function (IRF, orange squares) and fitted line (Fit, red) are also shown. **d, e** Comparison of absorption (**d**) and emission (**e**) spectra between RhGln (red line, 0.83 μM) and its rhodamine precursor (Rh101, blue line, 0.94 μM) at an $\lambda_{ex}$ of 530 nm in intracellular solution at 20 °C is shown with a relative fluorescence emission quantum yield value ($\Phi_F$) of 0.6 for RhGln (inset).

inhibited by blocking neuronal glutamine transport with methylaminoisobutyric acid (MeAIB, Fig. 3a, d, e), we investigated whether RhGln entered synaptic compartments upon activity. Confocal microscopy indicated that the extended punctate labeling showed increased co-localization with VGlut1, a glutamatergic presynaptic marker (Fig. 4a, b). To unequivocally identify the cellular structure of the RhGln punctate labeling, we used super-resolution STED imaging of RhGln-filled hippocampal GFAP-eGFP-positive astrocytes and confirmed that these punctate structures were indeed presynaptic (Fig. 4c–e). Together, our findings identify an activity-dependent mobilization of astroglial glutamine from astroglial networks into adjacent presynaptic structures.

**Astroglial connexin 43 is expressed in perisynaptic astroglial processes and displays activity-dependent hemichannel opening.** To track down the mechanism involved in activity-dependent glutamine supply by astrocytes, we focused on astroglial proteins that have regulatory roles in both the astroglial network and at a perisynaptic level. Astroglial connexins (Cxs) have both GJ and hemichannel (HC) functions, mediating the transfer of small

molecules within and out of the astroglial network, respectively[26]. This unique property may drive the redistribution of glutamine from the astroglial network into perisynaptic compartments and then synapses. While astrocytes abundantly express both Cx30 and 43 isoforms, evidence suggests that only Cx43 regulates physiological synaptic activity via its HC function[26,27]. However, to establish the involvement of Cx43 in glutamine supply, two prerequisites have to be fulfilled: expression in close proximity to synapses, and activity-dependent regulation of HC activity. To assess Cx43 localization, we used immunolabeling and found that Cx43 protein is expressed throughout hippocampal astrocytes including fine processes away from the cell soma and close to VGlut1-positive glutamatergic presynapses (Fig. 5a). We found that 38.2% of total Cx43 co-localizes with VGlut1 puncta (Fig. 5b). To further support this localization, we co-purified perisynaptic astroglial processes with hippocampal synaptosomes[28,29] and found an enrichment of Cx43 proteins in synaptosomal fractions compared to total hippocampal extracts (Fig. 5c, d). Preparations obtained from glial conditional Cx43 knockout mice (−/−) lacked Cx43 protein expression, confirming that this protein enrichment is of glial origin. Last, at the ultrastructural electron microscopic

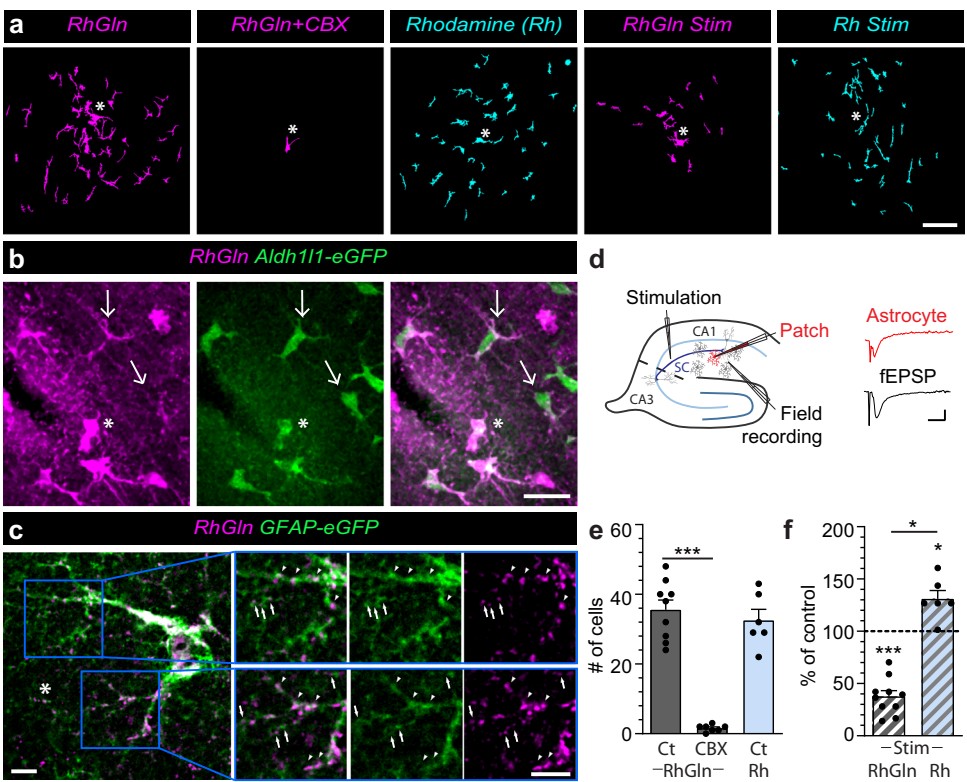

**Fig. 2 RhGln reveals activity-dependent redistribution of glutamine away from astroglial networks. a** RhGln or Rh traffics through gap junction-mediated astroglial networks when dialyzed (0.8 mM, 20 min) into a single CA1 hippocampal astrocyte via the patch pipette. The representative images illustrate the transfer of RhGln (magenta) or Rh (cyan) from the patched astrocyte (asterisks) to neighboring cells under control condition, which was abolished by the gap-junction blocker carbenoxolone (CBX, 200 μM), and reduced by repetitive synaptic stimulation (Stim; 10 Hz, 30 s every 3 min for 20 min). The transfer of Rh was similar to RhGln in basal condition, but was not reduced by stimulation. **b** RhGln under control conditions was found distributed in neighboring cells identified as Aldh1l1-positive astrocytes (green), indicating the spread of glutamine into astroglial networks. Asterisks mark the position of the patched cell. Thin arrows mark astrocytes in which RhGln-labeling is found primarily in processes. **c** RhGln puncta were observed along main (arrowheads) and fine processes (arrows) of a GFAP-positive cell (green). Higher magnification of two areas (blue boxes) are also shown. **d** Schematic and sample traces showing simultaneous recordings of evoked depolarization of a patched astrocyte (Patch, red trace) and field excitatory postsynaptic potential (fEPSP, Field recording, black trace) during stimulation of the Schaffer collaterals (Stimulation). **e** Quantification of RhGln or Rh transfer from the patched astrocyte to neighboring cells under different conditions (Control, Ct-RhGln, $n = 9$; CBX-RhGln, $n = 7$, $p < 0.0001$; Ct-Rh, $n = 6$, $p = 0.7745$; one-way ANOVA with Bonferroni's post hoc test with Ct-RhGln). **f** Quantification of RhGln or Rh transfer from the patched astrocyte to neighboring cells under stimulated conditions normalized to control (Stim-RhGln, $n = 10$, $p < 0.0001$; Stim-Rh, $n = 6$, $p = 0.0123$; one-sample $t$-test with 100%; $p = 0.0421$ two-tailed unpaired Student's $t$-test between Stim-RhGln and Stim-Rh). Mean ± SEM in **e**, **f**. Scale bars: **a** 50 μm; **b** 20 μm; **c** 5 μm; **d** 0.5 mV (Astrocyte), 0.2 mV (fEPSP), 20 ms. Asterisks indicate statistical significance (***$p < 0.0001$, *$p < 0.05$). Representative images in **a**, **b** are from replicates described in **e** and **f**; and **c** are from $n = 3$ replicates. Source data are provided as a Source data file.

level, we observed immunogold particles labeling Cx43 in small astroglial processes close to synaptic complexes. We found that 40.6% of total immunogold particles labeling Cx43 were present on astrocytic processes in close proximity to synaptic complexes (<300 nm away), which is similar to what was observed using immunolabeling (Fig. 5b). Distance analysis further revealed that most of these Cx43 positive gold grains were at a distance of 200–300 nm to the synaptic cleft, with a minimum distance of 72.2 nm and an average distance of 265.8 ± 115.5 nm to the nearest active zone (Fig. 5e, f).

To test whether physiological synaptic stimulation alters Cx43 HC activity, we performed ethidium bromide (EtBr) uptake assays in acute slices, and observed that synaptic stimulation (10 Hz, 30 s) markedly increased EtBr uptake (Fig. 6a–c). This enhanced uptake by stimulation was mediated by Cx43 HCs, as it was abolished by the Gap26 peptide, a specific blocker of Cx43 HCs, but not by a scrambled peptide (Gap26Sr, Fig. 6a–c). In addition, we observed that the Gap26 peptide alone decreased EtBr uptake significantly below control condition, which is in line with our previous findings showing that Cx43 HCs are open in

basal condition[27]. To elucidate the signaling mechanism involved in the activity-dependent opening of Cx43 HC mediating glutamine release, we focused on the most prominent candidates which are elevated during excitatory synaptic activity, namely glutamate[30] and potassium[31,32]. We found that the activity-dependent opening of astroglial Cx43 HC, induced by Schaffer collateral stimulation and assessed by EtBr uptake, was mediated by activation of ionotropic glutamate receptors, abundantly expressed in hippocampal neurons, as it was fully inhibited by antagonists of AMPARs (2,3-Dihydroxy-6-nitro-7-sulfamoyl-benzo[f]chinoxalin-2,3-dion, NBQX, 10 μM) and NMDARs (3-((R)-2-carboxypiperazin-4-yl)-propyl-1-phosphonic acid, CPP, 10 μM), but not of metabotropic glutamate receptors (LY341195, 20 μM, Fig. 7a, b). We also found that potassium channels played a critical role in the activity-dependent opening of Cx43 HC, since the potassium channel blocker $BaCl_2$ (200 μM) abolished the enhanced EtBr uptake upon stimulation. The potassium effect was specifically mediated by astroglial Kir4.1 channels, as evidenced by the lack of activity-dependent EtBr uptake in astrocytes deficient for Kir4.1. These findings suggest

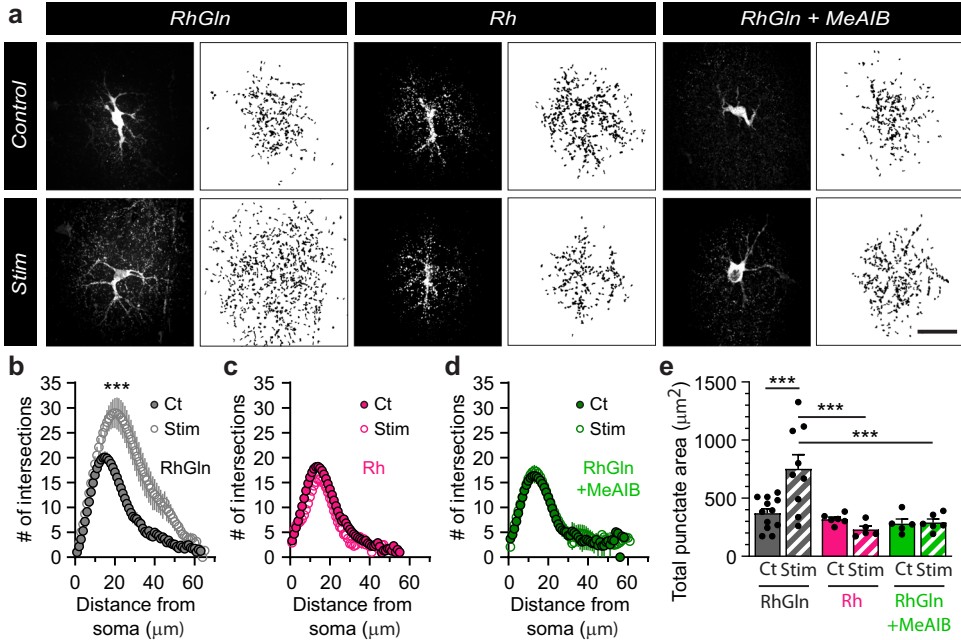

**Fig. 3 RhGln is redistributed into subcellular punctate structures. a** RhGln or Rh were dialyzed (0.8 mM, 20 min) into a single CA1 hippocampal astrocyte via the patch pipette. Punctate RhGln-labeling surrounding dye-filled astrocytes is enhanced by stimulation as shown in representative confocal (dark background) and thresholded binary (white background) images (**a** left), quantification by Sholl analysis (Ct, n = 12; Stim, n = 9, p < 0.0001, two-way ANOVA in **b**) and total punctate area (p = 0.0003 between Ct and Stim, one-way ANOVA with Bonferroni's post hoc test in **e**). This was not observed when the unconjugated rhodamine dye was used (Rh, 0.8 mM in **a** (middle): Ct, n = 6; Stim, n = 5, p = 0.7012, two-way ANOVA in **c**; and p > 0.9999 between Ct and Stim with Rh, and p < 0.0001 between RhGln Stim and Rh Stim, one-way ANOVA with Bonferroni's post hoc test in **e**) or in the presence of MeAIB, α-Methylaminoisobutyric acid, an antagonist of neuronal glutamine uptake (RhGln + MeAIB, 20 mM in **a** (right): Ct, n = 5; Stim, n = 6, p > 0.9999, two-way ANOVA in **d**; p > 0.9999 between Ct and Stim with RhGln + MeAIB, and p = 0.0002 between RhGln Stim and RhGln + MeAIB Stim, one-way ANOVA with Bonferroni's post hoc test in **e**). Mean ± SEM in **b–e**. Scale bars: **a** 20 µm. Asterisks indicate statistical significance (***p < 0.0001). Source data are provided as a Source data file.

that glutamate and potassium both contribute to the activity-dependent Cx43 HC opening via activation of neuronal ionotropic glutamate receptors and astroglial Kir4.1 channels, respectively. To further and directly assess the involvement of glutamate and potassium in the activation of Cx43 HC, we incubated hippocampal slices with either potassium (2 mM) or glutamate (1 µM) during EtBr uptake, and found that both similarly activated Cx43 HC in a Gap26-sensitive manner (Fig. 7c, d). To determine the relationship between these pathways, we repeated these experiments in hippocampal slices from mice deficient for astroglial Kir4.1 channels, and found that neither potassium nor glutamate, exogenously applied alone, could increase astroglial EtBr uptake in these mice. These data indicate that both pathways are linked, with the glutamate effect being mediated by potassium activation of astroglial Kir4.1 channels. Altogether, these data reveal that astroglial Cx43 is located near synapses, and suggest that its HC function is enhanced by glutamatergic synaptic activity controlling levels of extracellular potassium activating astroglial Kir4.1 channels.

**Connexin 43 mediates activity-dependent transfer of glutamine from hippocampal astrocytes to synapses.** Next, we tested whether Cx43 HCs mediate glutamine supply from astrocytes to synapses in an activity-dependent manner. To do so, we either genetically disrupted Cx43 using glial conditional Cx43 knockout mice (−/−), or acutely inhibited Cx43 HC activity with Gap26 treatment. In astrocytes lacking Cx43 (−/−), the diffusion of RhGln was not only decreased under basal conditions, but also became insensitive to synaptic stimulation (Fig. 8a, b). The same effects were observed with the Gap26 but not with the Gap26Sr peptide, indicating that activity-dependent glutamine transfer

from astrocytes to synapses is mediated by Cx43 HCs (Fig. 8a, b). Quantitative analysis of RhGln-positive punctate structures also confirmed that Cx43 HC-mediated transfer of glutamine from astrocytes to synapses occurs under basal conditions and is significantly enhanced by synaptic stimulation (Fig. 8a–c). Remarkably, restoring in vivo Cx43 expression selectively in hippocampal astrocytes from −/− mice using recombinant adeno-associated viruses (rAAV) (Fig. 9a–c) fully rescued activity-dependent RhGln transfer (−/− Cx43 Rescue), while the GFP control virus (−/− GFP Control) had no effect (Fig. 9d–g), further supporting a direct functional role of Cx43 in astroglial glutamine transfer.

**Astroglial glutamine supply via Cx43 hemichannels is required for synaptic transmission and recognition memory.** With direct visual evidence of an activity- and Cx43-dependent mobilization of astroglial glutamine into synaptic structures, we next investigated the functional relevance of this process. To determine its contribution to the replenishment of the presynaptic pool of glutamate, we examined fEPSPs at hippocampal CA3 to CA1 synapses in response to 10 Hz repetitive stimulation of Schaffer collaterals in acute slices (Fig. 10a, b). Upon stimulation at 10 Hz, a fast synaptic facilitation followed by a depression, resulting from presynaptic glutamate depletion, was observed (Fig. 10b–d). We found that, compared to control +/+ slices, excitatory synaptic transmission depressed faster in −/− slices as well as in +/+ slices treated with Gap26, but not with Gap26Sr (Fig. 10c–e). This indicates that Cx43 HCs are crucial for sustaining repetitive excitatory synaptic activity. We then tested whether the impairment of excitatory synaptic transmission upon loss of Cx43 resulted from lack of an astroglial glutamine supply.

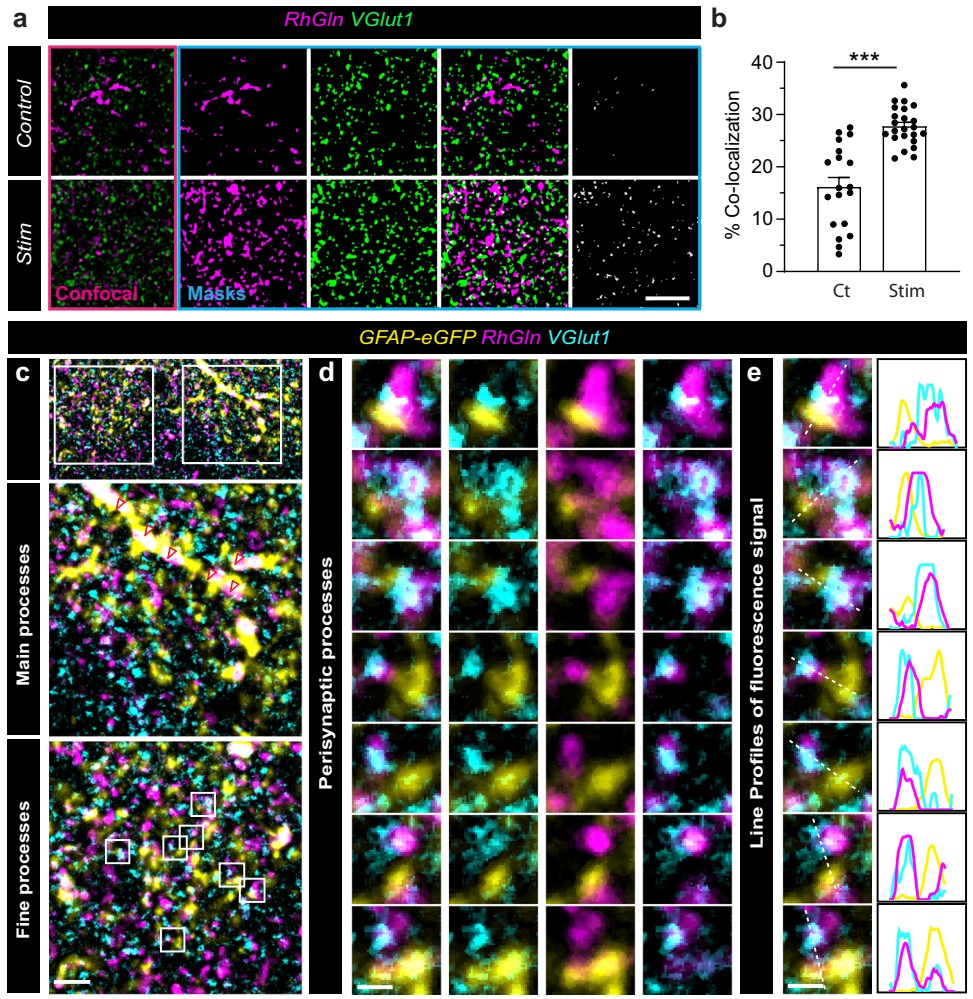

**Fig. 4 Astroglial RhGln enters presynaptic compartments upon synaptic activity. a** Sample confocal (pink box) and binary images (blue box) showing that after dialysis of a single astrocyte with RhGln (0.8 mM, 20 min), RhGln-labeled puncta (magenta) are observed and show increased co-localization with the glutamatergic presynaptic marker VGlut1 (green) under repetitive synaptic stimulation (Stim; 10 Hz, 30 s every 3 min for 20 min). **b** Bar graph (mean ± SEM) showing % co-localization normalized to total area of RhGln-filled structures for Ct (n = 19 fields, 3 independent experiments) and Stim (n = 23 fields, 3 independent experiments, p < 0.0001, two-tailed unpaired Student's t-test) conditions. **c, d** STED super-resolution imaging confirmed co-localization of VGlut1 (cyan) with RhGln (magenta) found directly next to an astroglial compartment (yellow) after dialysis of a GFAP-eGFP expressing astrocyte (yellow) with RhGln (0.8 mM, 20 min) during repetitive synaptic stimulation, as shown in the sample images (**c**). Higher magnifications of regions containing either main (middle panel) or fine (lower panel) astroglial processes are shown. Arrowheads denote accumulation of RhGln along a main process. **d** Representative images containing perisynaptic astroglial processes marked by boxes in **c** (lower panel). **e** Line profiles measured along individual white dotted lines for individual channels to illustrate co-localization. Scale bars: **a** 5 μm; **c** 2 μm; **d, e** 0.5 μm. Asterisks indicate statistical significance (***p < 0.0001). Representative images **a** are from replicates described in (**b**); and **c–e** are from n = 3 replicates. Source data are provided as a Source data file.

To do so, we treated acute hippocampal slices with exogenous glutamine (4 mM for 1–4 h), and found in these slices that Cx43 genetic disruption in astrocytes (−/−) or acute pharmacological Gap26 treatment no longer impaired excitatory transmission upon repetitive stimulation (Fig. 10f, g). Thus, exogenous glutamine fully rescued repetitive synaptic activity to wild-type levels. Importantly, the same exogenous glutamine treatment in +/+ slices had no effect on excitatory transmission (Fig. 10h), indicating that endogenous glutamine level in +/+ slices is sufficient to sustain evoked synaptic activity under physiological conditions. Furthermore, either Cx43 genetic disruption in astrocytes (−/−) or acute pharmacological inhibition of Cx43 HC by Gap26 in +/+ slices also led to an impairment in basal synaptic transmission, evoked by a single Schaffer collateral stimulation of low intensity (10–20 μA, 0.1 ms), as revealed by fEPSP slope/FV amplitude (Supplementary Fig. 5). Similar to the results obtained

with repetitive stimulation, exogenous glutamine (4 mM for 1–4 h) also fully rescued the basal synaptic transmission in +/+ slices treated with Gap26 or in −/− slices (Supplementary Fig. 5). Altogether, these findings indicate that Cx43 HCs contribute to physiological excitatory synaptic transmission by fueling synapses with glutamine. The hippocampus is well-known for its functions in learning and memory, including recognition memory[33]. Because the presynaptic pool of glutamate contributes to recognition memory[34], we investigated in vivo the functional significance of Cx43 HC-mediated release of glutamine in such form of hippocampal memory. To do so, we administered intra-hippocampal injections of Gap26 or control Gap26Sr to adult wild-type mice, then subjected them to a novel object recognition test (Fig. 10i). As mice are naturally attracted towards novelty, recognition memory was assessed by the relative time spent exploring a novel versus familiar object. While control Gap26Sr-

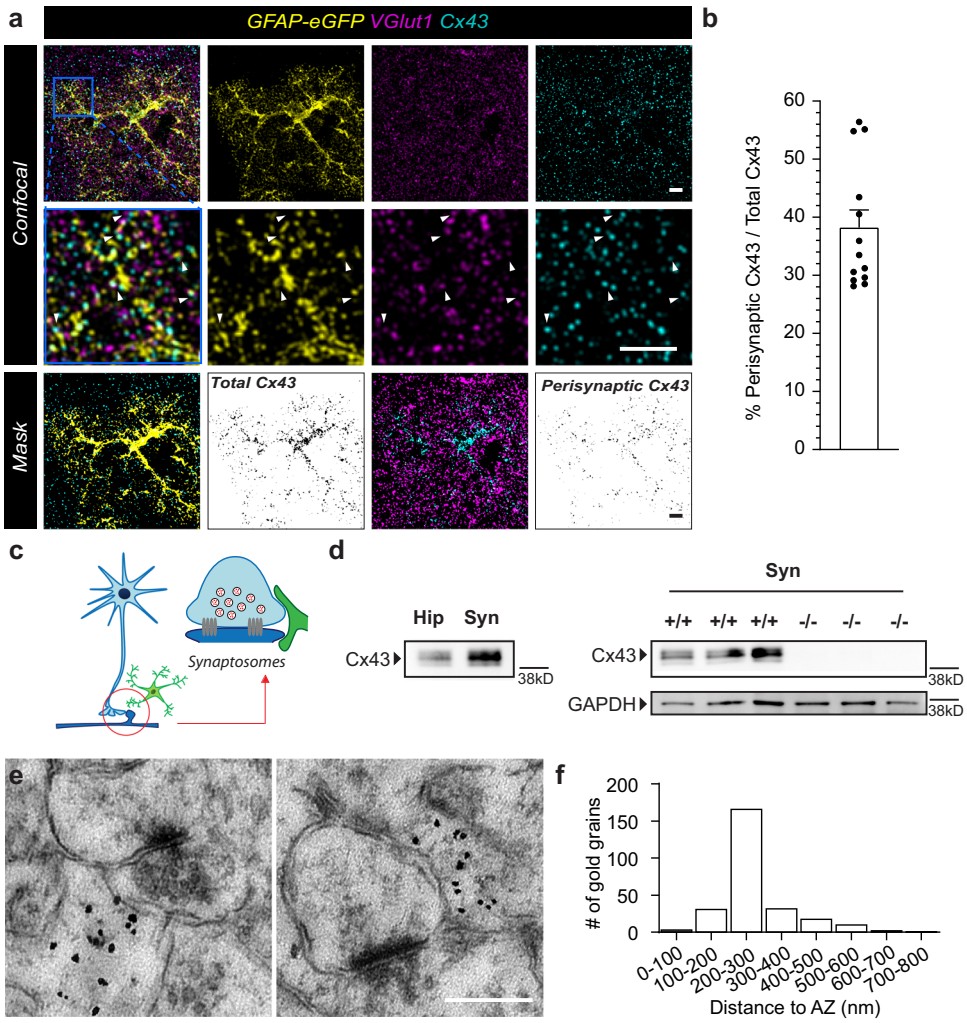

**Fig. 5 Cx43 is expressed in perisynaptic astroglial processes. a** Representative confocal images showing close proximity of Cx43 (cyan) in GFAP-eGFP positive astrocytes (yellow) to presynaptic structures immunolabeled for VGlut1 (magenta). Higher magnification images of a region containing astroglial processes (blue square) are shown in the middle row. Arrowheads denote points of close contact. Masks showing co-localized area of GFAP (yellow) and Cx43 (cyan) as total Cx43 (binary inverse image) and co-localized area of total Cx43 (cyan) and VGlut1(magenta) as presynaptic Cx43 (binary inverse image). **b** Bar graph (mean ± SEM) showing % Perisynaptic Cx43 normalized to total Cx43 area ($n = 13$ fields, 3 independent experiments). **c** Schematic illustration of co-purification of perisynaptic astroglial processes in crude synaptosomes. **d** Representative western blots showing an enrichment of Cx43 protein in synaptosomal preparations (Syn) compared to total hippocampal lysates (Hip) in wild type ($+/+$), but not in glial conditional Cx43 knockout ($-/-$) mice. GAPDH was used as a loading control. **e** Representative high magnification electron micrographs showing the presence of Cx43 protein labeled by immunogold particles in astroglial processes near synaptic complexes. **f** Distribution histogram of distance between Cx43 gold grains and the nearest active zone. Scale bars: **a** 5 μm, **e** 0.5 μm (left), 0.3 μm (right). Representative images (**a**) are from replicates described in (**b**); **d** are from $n = 3$ replicates; and **e** are from $n = 8$ replicates. Source data are provided as a Source data file.

injected mice spent significantly more time exploring the novel object, this preference was not present in Gap26-treated mice, indicating impaired recognition memory (Fig. 10j). Remarkably, we found that mice co-injected with Gap26 and glutamine no longer exhibited memory impairment, implying that exogenous glutamine fully rescued object recognition (Fig. 10j), as it did for synaptic transmission (Fig. 10f, g). Our data thus show that the Cx43HC-dependent regulation of hippocampal excitatory synaptic transmission and recognition memory are both mediated by glutamine.

## Discussion
In this study, we developed a fluorescent glutamine molecule to directly track glutamine in live cells. Using this probe, we provide direct evidence for activity-dependent redistribution and supply of glutamine from perisynaptic astrocytes to synaptic structures under physiological conditions. Furthermore, we identified astroglial Cx43 HCs as the key mediators of this process, essential for sustaining physiological excitatory synaptic transmission as well as recognition memory. Our findings thus address a long-standing controversy and directly demonstrate a requirement for astroglial glutamine in supporting both physiological synaptic transmission and cognition via a previously unknown mechanism mediated by Cx43 HCs.

**A glutamine fluorescent probe to assess astroglial glutamine supply in situ in live cells.** Up to now, the occurrence and role of astroglial glutamine supply during physiological processes has lacked support from tools that can directly evaluate glutamine cellular transfer. Initially, radiolabeled-glutamine and electron microscopy of glutamine immunogold labeling enabled quantification of brain glutamine levels[16,17] and glutamine subcellular

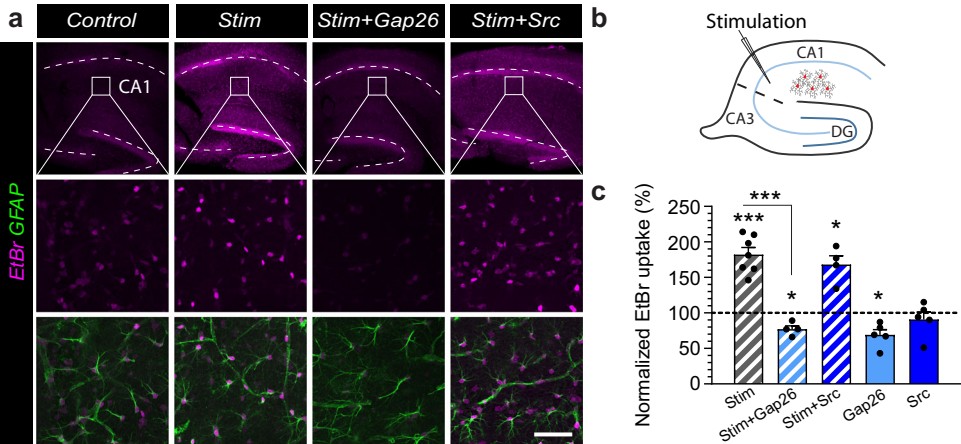

**Fig. 6 Cx43 hemichannel function is enhanced by synaptic activity.** Acute hippocampal slices were loaded with ethidium bromide (EtBr) under different experimental conditions. **a** Sample images of EtBr uptake (magenta) in hippocampal GFAP-immunolabeled astrocytes (green) are shown for Control, Stim (10 Hz, 30 s every 3 min for 20 min) in the absence or presence of the Cx43 HC blocker Gap26 or a Gap26 scramble version (Src). Higher magnifications of the CA1 stratum radiatum subregion are shown in bottom two rows. **b** Schematic illustrating stimulation of hippocampal Schaffer collaterals and EtBr uptake in neighboring astrocytes. **c** Quantification of EtBr uptake normalized to 100% control (dotted line) is shown. Stimulation-enhanced EtBr uptake by nearly 2-fold (mean ± SEM; Control, $n = 6$; Stim, $n = 7$, $p = 0.0002$ between Control and Stim, one-sampled $t$-test). This enhanced uptake was not observed in the presence of Gap26 (Stim + Gap26, $n = 4$, $p < 0.0001$ between Stim and Stim+Gap26, one-way ANOVA with Bonferroni's post hoc test; $p = 0.0168$ with control, one-sampled $t$-test) but persisted with Src (Stim + Src, $n = 4$, $p = 0.6857$ between Stim and Stim+Src, one-way ANOVA with Bonferroni's post hoc test; $p = 0.0125$ with control, one-sampled $t$-test), while Gap26 alone decreases EtBr uptake from control level ($n = 5$, $p = 0.014$, one-sampled $t$-test) but not Src ($n = 5$, $p = 0.4443$, one-sampled $t$-test). Scale bars: **a** 450 μm (top), 50 μm (middle and bottom). Asterisks indicate statistical significance (***$p < 0.001$; *$p < 0.05$). Source data are provided as a Source data file.

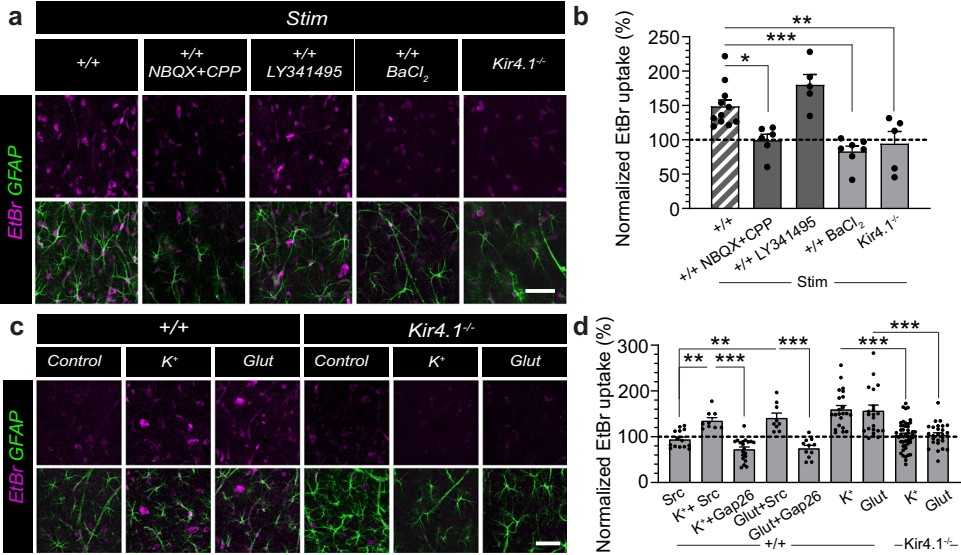

**Fig. 7 Activity-dependent Cx43 hemichannel function is mediated by glutamatergic synaptic activity and potassium activation of Kir4.1 channels.** Acute hippocampal slices were loaded with ethidium bromide (EtBr) under stimulated conditions in the absence or presence of various blockers. **a**, **b** Sample images of EtBr uptake (magenta) in hippocampal GFAP-immunolabeled astrocytes (green) are shown in (**a**) and quantified in (**b**). Stimulation-enhanced uptake was blocked in the presence of NBQX + CPP (20 μM, $n = 6$, $p = 0.0112$ with wild-type control, +/+, $n = 11$, one-way ANOVA with Bonferroni's post hoc test), but not LY341495 (20 μM; $n = 5$, $p = 0.2433$ with +/+, one-way ANOVA with Bonferroni's post hoc test), indicating the specific involvement of ionotropic glutamate receptor activity. This stimulation-dependent uptake was also blocked in the presence of BaCl$_2$ (200 μM; $n = 7$, $p = 0.0004$ with +/+, one-way ANOVA with Bonferroni's post hoc test) and in acute slices prepared from Kir4.1−/− mice ($n = 5$, $p = 0.0086$ with +/+, one-way ANOVA with Bonferroni's post hoc test), indicating the specific involvement of Kir4.1 activity. **c**, **d** Incubation of acute slices with either 2 mM K$^+$ or 1 μM glutamate (Glut) alone enhanced basal EtBr uptake (K$^+$ + Src, $n = 9$, 3 experiments, $p = 0.0061$; Glut+Src, $n = 9$, 3 experiments, $p = 0.0056$ with Src, $n = 14$, 5 experiments; Kruskal–Wallis test with Dunn's post hoc test) in a Cx43 HC-dependent manner (K$^+$ + Gap26, $n = 20$, 7 experiments, $p < 0.0001$ with K$^+$ + Src; Glut+Gap26, $n = 11$, 5 experiments, $p < 0.0001$ with Glut+Src; Kruskal–Wallis test with Dunn's post hoc test). In Kir4.1−/− hippocampal slices, neither K$^+$ nor Glut were able to enhance EtBr uptake (K$^+$ Kir4.1−/−, $n = 49$, 17 experiments, $p < 0.0001$ with K$^+$ +/+, $n = 22$, 8 experiments; Glut Kir4.1−/−, $n = 27$, 9 experiments, $p = 0.0001$ with Glut +/+, $n = 23$, 8 experiments, Mann–Whitney test). Mean ± SEM in (**b**) and (**d**). Scale bars: **a** and **c**, 50 μm. Asterisks indicate statistical significance (***$p < 0.001$; **$p < 0.01$; *$p < 0.05$). NBQX = 2,3-Dihydroxy-6-nitro-7-sulfamoyl-benzo[f]chinoxalin-2,3-dion; CPP = (3-((R)-2-carboxypiperazin-4-yl)-propyl-1-phosphonic acid. Source data are provided as a Source data file.

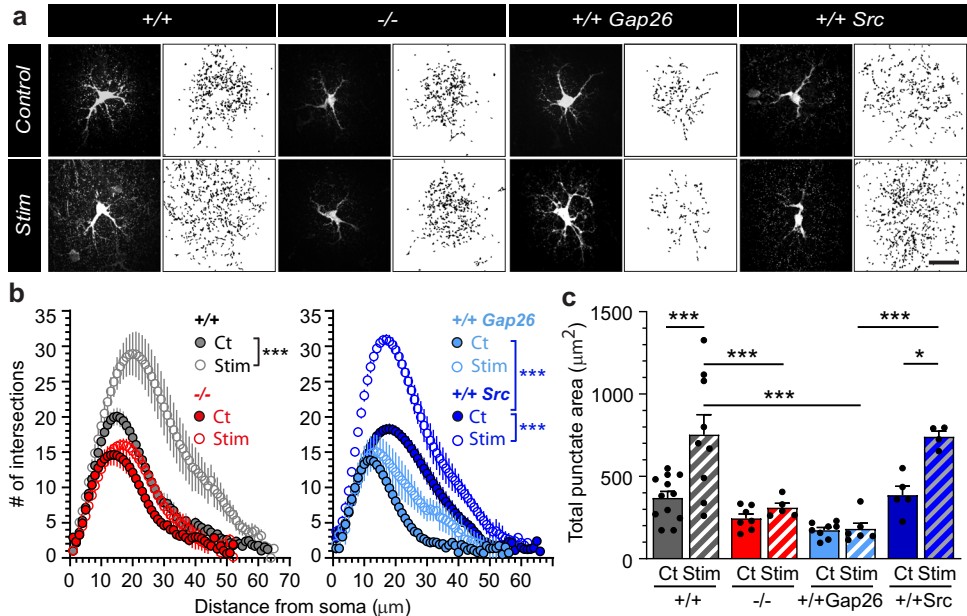

**Fig. 8 Cx43 hemichannels mediate activity-dependent transfer of glutamine from astrocytes to synapses. a** Representative confocal (dark background) and thresholded binary (white background) images of hippocampal CA1 astrocytes dialyzed with RhGln (0.8 mM, 20 min) via the patch pipette under control or stimulated (10 Hz, 30 s every 3 min for 20 min) conditions in acute slices obtained from: wild-type mice (+/+), glial conditional Cx43 knockout mice (−/−) or wild-type mice exposed to Gap26 (+/+ Gap26) or Gap26 scramble (+/+Src) peptides. The binary images were quantified by Sholl analysis (**b**) and total punctate area (**c**). In +/+ mice, repetitive synaptic stimulation strongly increased the punctate RhGln-labeling compared to control as shown by both Sholl analysis (+/+: Ct, $n = 12$; Stim, $n = 9$, $p < 0.0001$, two-way ANOVA in **b**) and total punctate area ($p < 0.0001$ between Ct and Stim in +/+, one-way ANOVA with Bonferroni's post hoc test in **c**). This was abolished in −/− mice (−/−: Ct, $n = 7$; Stim, $n = 5$, $p = 0.0826$, two-way ANOVA in **b**; $p > 0.999$ between Ct and Stim in −/−, and $p = 0.0002$ between +/+ Stim and −/− Stim, one-way ANOVA with Bonferroni's post hoc test in **c**) or in the presence of Gap26 (+/+ Gap26: Ct, $n = 8$; Stim, $n = 5$, $p = 0.0651$, two-way ANOVA in **b**; $p > 0.999$ between Ct and Stim in +/+ Gap26, and $p < 0.0001$ between Stim of +/+ and +/+ Gap26, one-way ANOVA with Bonferroni's post hoc test in **c**), but unchanged in the presence of the scramble Gap26 peptide (+/+Src: Ct, $n = 5$; Stim, $n = 4$, $p < 0.0001$, two-way ANOVA in **b**; $p = 0.025$ between Ct and Stim with +/+ Src, and $p < 0.0001$ between Stim of +/+ Gap26 and +/+ Src, one-way ANOVA with Bonferroni's post hoc test in **c**). Gap26 alone also inhibited the spread of glutamine, suggesting a basal transfer of glutamine into synaptic structures which is dependent on Cx43 HC activity ($p < 0.0001$ between +/+ Gap26 Ct and +/+ Gap26 Stim, two-way ANOVA in **b**). Mean ± SEM in (**b**), (**c**). Scale bar: **a** 20 μm. Asterisks indicate statistical significance (***$p < 0.001$; *$p < 0.05$). Source data are provided as a Source data file.

localization[35], respectively. However, these approaches cannot provide dynamic information about changes in glutamine content. More recently, FRET-based biosensors have been reported to dynamically monitor intracellular levels glutamine[18,19]. However, these genetically-introduced sensors lack the ability to provide spatially-resolved directionality of glutamine flow between different cell types. Similar to the use of a fluorescently tagged glucose[36] designed to probe local and fast brain metabolism, here we have taken a direct approach to track cellular glutamine transfer in live cells in situ using fluorescently tagged glutamine molecules.

Our probe, RhGln, exhibits several key features required for live detection of glutamine in cellular compartments. Its water solubility permits use in physiological extracellular and intracellular milieu. Its bright fluorescence allows clear detection in small processes using super-resolution imaging. Its small size of 0.64 kDa enables intracellular and extracellular diffusion through small transmembrane channels such as Cx HCs. Finally, its stability as a non-hydrolysable glutamine analog enables detection of glutamine rather than its metabolites. We have indeed engineered a fluorescent tag on the amide side chain at an optimal distance from glutamine. As a result, this conjugation prevents conversion of glutamine to glutamate, while leaving the glutamine function intact for its biological functions. As glutamine is readily metabolized intracellularly, this aspect of our probe is essential to ensure unambiguous visualization of glutamine mobilization. It also allows us to observe the accumulation of glutamine at its

site of storage or conversion. This makes RhGln a useful tool to study immediate mobilization of glutamine as its unmetabolized form. It is however not intended to be used to trace further glutamine metabolism or to interfere with excitatory synaptic transmission.

Using this probe, we show that upon physiological synaptic activity, RhGln that has been selectively delivered to single astrocytes via patch-clamp redistributes from the astroglial network into neighboring presynaptic structures. We tracked the probe at a subcellular level, revealing activity-dependent punctate labeling surrounding the RhGln-injected astrocyte, that was not associated with changes in astroglial morphology, but was localized to presynaptic structures, as revealed by super-resolution STED microscopy. This finding demonstrates the occurrence of a directional glutamine cellular transfer from astrocytes to presynaptic elements under physiological conditions. Of note, the activity-dependent restriction of intracellular dye transfer into neighboring astrocytes was unique to RhGln, and not observed using the Rh dye alone. In addition, the specific RhGln spatial pattern displayed upon activity, consisting of punctate labeling of presynaptic compartments, was also not observed with the unconjugated Rh dye alone, and was inhibited upon blockade of neuronal glutamine transport. This indicates that upon activity, RhGln enters synaptic compartments via neuronal glutamine transport, in contrast to the inert control dye, and thus retains the specificity of glutamine cellular trafficking. Our probe therefore offers an advantage over existing methods

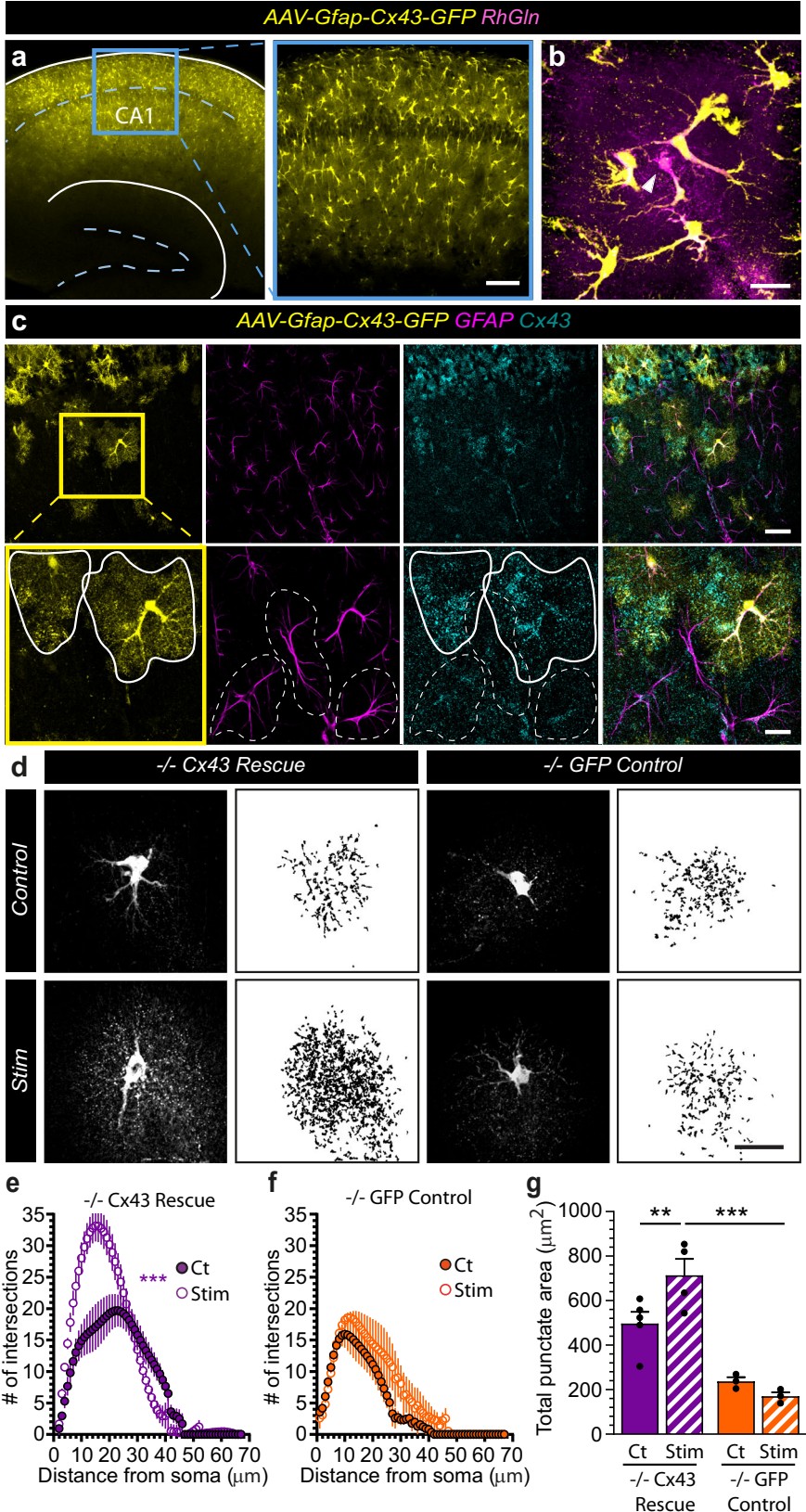

and is uniquely suitable for investigating directional transfer of glutamine in live cells. Its use in hippocampal tissues here provides a direct visual evidence of an activity-dependent mobilization of astroglial glutamine into presynaptic terminals in physiological conditions in situ.

**Astroglial glutamine supply sustains physiological activity.** The pool of presynaptic glutamate is limited. According to estimates, this pool would be exhausted within minutes of physiological synaptic activity without replenishment. This estimate held true at both small presynaptic terminals from hippocampal CA3 to

**Fig. 9 Restoring in vivo Cx43 expression in hippocampal astrocytes from Cx43−/− mice rescues activity-dependent transfer of glutamine. a** Sample image of the hippocampus of Cx43−/− mice injected intra-hippocampally with rAAV2/9-GFAP-Cx43-GFP virus, showing numerous cells expressing Cx43-GFP in the CA1 area. The blue box is magnified on the right. **b** RhGln (magenta) was loaded into a Cx43-GFP expressing astrocyte (yellow) via a patch pipette as shown. Arrow head indicates the patched cell. **c** Sample images after immunostaining showing specific expression of Cx43 (cyan) in Cx43-GFP-positive (yellow) astrocytes (GFAP, magenta). The yellow box is magnified in the bottom row. Solid and dotted white lines outline GFP-positive and -negative astrocytes, respectively. **d**–**g** Cx43−/− mice first received either rAAV2/9-GFAP-Cx43-GFP (−/− Cx43 Rescue, **d** left, **e** and **g**) or rAAV2/9-GFAP-GFP (−/− GFP Control, **d** right, **f** and **g**) virus. Hippocampal astrocytes were then dialyzed with RhGln under either control or synaptic stimulation (10 Hz, 30 s) conditions for 20 min. Both representative confocal (dark background) and thresholded binary (white background) images are shown in (**d**) for each condition. The binary images were quantified by Sholl analysis (**e**, **f**) and total punctate area (**g**). The stimulation-induced transfer of RhGln was rescued in Cx43−/− mice ($n = 5$) by restoring Cx43 expression selectively in astrocytes via viral infection shown by both Sholl analysis (−/− Cx43 rescue, $n = 4$, $p < 0.0001$, two-way ANOVA for **e**) and total punctate area ($p = 0.0031$ between Ct and Stim with −/− Cx43 Rescue, one-way ANOVA with Bonferroni's post hoc test for **g**) as compared to the GFP control infection (−/− GFP Ct, $n = 3$; Stim, $n = 3$, $p = 0.9856$, two-way ANOVA for **f**, $p > 0.999$ between Ct and Stim with −/−GFP Control and $p < 0.0001$ between Stim of −/− Cx43 Rescue and −/− GFP Control, one-way ANOVA with Bonferroni's post hoc test for **g**). Mean ± SEM in (**e**)–(**g**). Scale bars: **a** 200 μm (left) and 50 μm (right); **b** 50 μm (top) and 20 μm (bottom); **c**, **d** 20 μm. Asterisks indicate statistical significance (***$p < 0.001$; **$p < 0.01$). Representative images **a**, **b** from replicates described in (**g**); **c** are from $n = 3$ replicates. Source data are provided as a Source data file.

CA1 synapse and larger synapses at the calyx of Held synapse[6]. The mechanism by which astrocytes could locally replenish the presynaptic pool of glutamate has thus been seen as a suitable pathway to sustain brain activity in the absence of glutamine synthesis at excitatory synapses[4,5]. Astrocytes are indeed a well-established source of recycled glutamate. Their numerous peri-synaptic processes, strongly enriched in glutamate transporters[2,37], provide efficient glutamate clearance upon synaptic activity, while their specific expression of glutamine synthetase enables the conversion of sequestered glutamate into glutamine[3]. Surprisingly, despite the need of rapid glutamate recycling for normal neurotransmission, until now the significance of the glutamate–glutamine cycle has only been clearly evident during high glutamate turnover, for example epileptiform activity[7–9]. The occurrence and relevance of this cycle under physiological conditions remains controversial, not only due to lack of tools to directly track glutamine transfer, but also due to conflicting findings on neuronal activity. Previous work in cortical and hippocampal neurons in vitro[11,38], hippocampal slices[10], retina[12], and in vivo[13] suggested that the glutamate–glutamine cycle is not required for physiological synaptic transmission, network activity, or memory. In contrast, other studies performed in brainstem slices[14], isolated excitatory terminals from hippocampal slices or active neocortical slices[15] reported a role for presynaptic glutamine transport and glutamine synthesis, respectively, in supporting glutamatergic synaptic transmission. These contrasting findings likely stem from different experimental paradigms, including the use of distinct tools to interfere with the cycle, variations in preparations, brain regions, and types of synapses, which are not directly comparable due to disparate presynaptic pool size, release probability, and thus variation in glutamine requirement. Furthermore, these studies mostly relied on pharmacological blockers to perturb different components of the cycle, and may thus suffer from incomplete inhibition as well as significant off-target effects.

Here, using a multidisciplinary approach combining chemical biology to synthetize a fluorescent glutamine probe, super-resolution imaging, electrophysiology, and behavioral testing, we demonstrate a requirement for astroglial glutamine in sustaining physiological synaptic and cognitive demands. We uncovered a significant glutamine-dependent component of excitatory transmission evoked by synaptic stimulation at 10 Hz, a frequency within the physiological range for hippocampal CA3 to CA1 synapses[39], known to initially facilitate glutamate release and then to deplete presynaptic pools[40]. This glutamine component was essential not only for the initial facilitation of excitatory synaptic transmission, but also for its maintenance,

indicating its crucial role in the replenishment of the presynaptic pool of glutamate. This dependency on Cx43HC and glutamine was also observed for basal synaptic transmission in response to a single Schaffer collateral stimulation, consolidating the role of glutamine in physiological synaptic transmission. We showed that astrocytes are the source of glutamine, and supply is dependent on Cx43, a gap junction channel subunit abundantly expressed in astrocytes. Using imaging of the RhGln probe, we indeed found that astrocytes, via Cx43, mediate an activity-dependent glutamine supply from astrocytes to synapses. Furthermore, our electrophysiological recordings reveal that excitatory transmission evoked synaptically by the same regime of activity also relies on astrocytes, via Cx43, and glutamine supply. We indeed found that exogenous glutamine fully rescued the impaired excitatory transmission evoked synaptically by either a single or repetitive stimulation in mice lacking Cx43 expression conditionally in astrocytes. Importantly, we also found that endogenous levels of glutamine are not only required, but sufficient to sustain evoked excitatory synaptic transmission since exogenously supplying glutamine in wild-type mice had no influence on glutamatergic synaptic transmission. It has been previously reported that glutamine uptake by neuronal glutamine transporters in transfected fibroblasts is saturable and highly sensitive to glutamine with a Michaelis constant (Km) of $489 \pm 88$ μM at pH 7.4[41]. The authors showed that transporter saturation is reached in the presence of 1 mM glutamine. Based on this, and considering slower diffusion in brain tissues compared to cultured cells, the addition of 4 mM of glutamine exogenously on acute hippocampal slices is likely to saturate neuronal glutamine transporters, assuring sufficient rescue. Thus, we provide direct quantitative evidence that evoked excitatory synaptic transmission at hippocampal CA3 to CA1 synapses relies on astroglial glutamine.

**Cx43 hemichannels as a molecular pathway for astroglial glutamine supply to synapses.** Cxs in astrocytes form both GJs and HCs, and underlie the transfer of small molecules within and out of the astroglial network, respectively[42]. GJs, by mediating the extensive network communication of astrocytes, contribute to several key physiological functions, including metabolic support and extracellular homeostasis[42]. In contrast, Cx43 HCs are thought to be active mostly in pathological conditions, where they contribute to pathophysiological processes via release of molecules such as ATP or prostaglandins[43]. Here we reveal that Cx43 HCs are not only active under physiological conditions, but also sustain synaptic transmission and cognition through glutamine release. Although Cx43 GJs have been shown to be permeable to

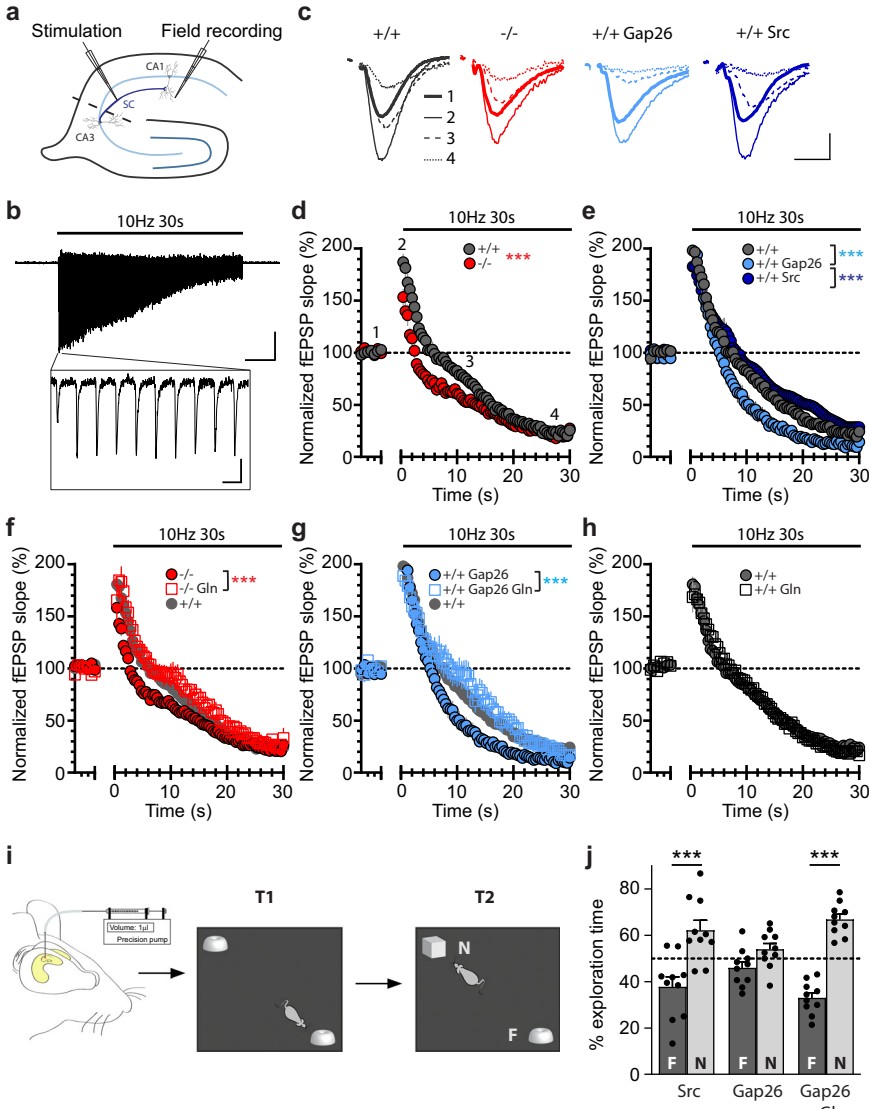

**Fig. 10 Astroglial glutamine supply via Cx43 hemichannels sustains glutamatergic synaptic transmission and is required for novel object recognition memory. a** Schematics depicting recording of field excitatory postsynaptic potentials (fEPSP, field recording) evoked by Schaffer collaterals (SC) stimulation in the CA1 region of hippocampal slices. **b** Representative trace for fEPSP recorded during stimulation (10 Hz, 30 s) is presented with the first 10 responses magnified in inset. **c** Sample traces are shown for fEPSPs recorded before (1), at the start of (2), during (3), and at the end of (4) the stimulation. **d, e** Plots of fEPSP slope normalized to baseline upon 10 Hz stimulation show a decrease in synaptic transmission in slices from −/− (**d**, $n = 11$, $p < 0.0001$) or with Gap26 treatment (**e**, +/+Gap26, $n = 9$, $p < 0.0001$) as compared to +/+ ($n = 11$ for **d**, $n = 14$ for **e**), whereas there was no effect of the scramble Gap26 treatment (**e**, +/+Scr, $n = 6$, $p = 0.9994$ with +/+, $p < 0.0001$ with +/+Gap26) (two-way ANOVA). Numbers 1–4 in d correspond to regions referred to in (**c**). **f–g** Glutamine pre-treatment (4 mM, 1–4 h) in −/− slices (**f**, $n = 16$ for −/−, $n = 8$ for −/−Gln, $p < 0.0001$) and Gap26-treated slices (**g**, $n = 9$ for +/+Gap26, $n = 7$ for +/+Gap26 Gln, $p < 0.0001$) rescued transmission to +/+ levels (two-way ANOVA). +/+ plot is shown for comparison. **h** Glutamine pre-treatment alone in +/+ mice had no effect ($n = 16$ for +/+, $n = 8$ for +/+Gln, $p > 0.9999$, two-way ANOVA). **i** Mice underwent intra-hippocampal injections of either Gap26 scramble (1 mM), Gap26 (1 mM), or Gap26 (1 mM)+Glutamine (200 mM) 30 min before being submitted to the novel object recognition task which consisted of first an acquisition trial (T1, exploration of 2 similar objects) and then 24 h later to a restitution trial (T2, exploration of a familiar object "F" from T1 and a novel object "N"). **j** Percent exploration time revealed a loss of preference for the novel object in mice treated with Gap26, but was rescued by co-injection of glutamine (Gap26 + Gln). $n = 10$ in each condition ($p < 0.0001$ between F and N with Src, $p = 0.2368$ between F and N with Gap26, $p < 0.0001$ between F and N with Gap26+Gln, two-way ANOVA with Bonferroni's post hoc test). Mean ± SEM in (**d–h**) and (**j**). Scale bars: **b** 0.2 mV, 5 s (full trace), 100 ms (inset); **c** 0.2 mV, 10 ms. Asterisks indicate statistical significance (***$p < 0.001$). Source data are provided as a Source data file.

several metabolites such as glucose, lactate, or glutamine[23,44], until now this has not been the case for Cx43 HCs[45]. We thus identify astroglial Cx43 HCs as a release pathway for glutamine controlling physiological neuronal activity. Using immunohistochemistry, biochemistry, and electron microscopy, we first show that Cx43 is present near hippocampal synapses. We have

verified that ~40% of astroglial Cx43 is found near synapses. We also report functionally that Cx43 HCs open under physiological conditions in an activity-dependent manner. Here, we also showed that Cx43 HC are open during basal condition as previously reported[27]. Our observation that RhGln spread from astrocytes to synapses is also inhibited during basal condition

(Fig. 8b) suggests that there is a basal glutamine transfer from the astrocytes to the synapses. Based on our identification of the involvement of glutamatergic synaptic activity triggering Kir4.1-dependent opening of Cx43 HC upon synaptic stimulation, we postulate that an activity-dependent elevation of synaptic glutamate first act on ionotropic glutamate receptors from post-synaptic sites. The resulting release of potassium from postsynaptic elements following glutamatergic activity, known to represent the main source (80%) of neuronal potassium release[46], then in turn activates Kir4.1 channels at perisynaptic astroglial processes resulting in Cx43 HC opening. Next, using super-resolution imaging and electrophysiology, we found that Cx43 HC activity is essential for the activity-dependent transfer of glutamine from astrocytes to synapses, and that this HC-mediated glutamine supply is crucial for sustaining physiological excitatory synaptic activity. Rescuing Cx43 expression selectively in astrocytes from Cx43 conditional knockout mice indeed restored astroglial glutamine transfer. In addition, blocking Cx43 HC activity, either chronically and non-selectively using astroglial conditional knockout mice, or acutely using specific blocker peptides, impaired astroglial glutamine transfer and excitatory synaptic transmission. We demonstrated that glutamine is required for sustaining synaptic transmission by restoring normal activity with exogenous glutamine. It is noteworthy that a few studies recently reported that Cx43 HCs can regulate neuronal activity via different mechanisms, including synaptic activity and plasticity via release of ATP[27] or D-serine[47], or network activity, i.e. olfactory bulb slow oscillations[24] and hippocampal surround inhibition within synaptically active regions[48], both via ATP release.

We also report that astroglial glutamine supply via Cx43 HCs is relevant for cognition. We found that glutamine supply associated with Cx43 HC activity is essential for cognitive performance requiring hippocampal memory, as mice with acutely blocked Cx43 HCs, were no longer able to recognize a novel object in the presence of Gap26. This defect was rescued by concomitant in vivo delivery of exogenous glutamine. These data are consistent with a recent study indicating a role for Cx43 HCs in spatial short-term memory[49], although the underlying mechanism was not identified in this work. Our findings showing an involvement of glutamate in opening Cx43 HCs could explain a previous study showing that an increased glutamate efflux in the hippocampus is associated with the recognition of novel but not familiar objects[50].

It is noteworthy that almost two decades ago it was suggested that multiple pathways are responsible for astroglial glutamine efflux in vitro, and that astroglial glutamine transporters only account for ~50% of all glutamine release from astrocytes[51]. The authors attributed the remaining ~50% of basal glutamine efflux to yet unidentified mechanisms, however, pharmacologically characterized it as $Na^+$-independent. Incidentally, transmembrane channels such as Cx43 GJs can transport metabolites such as glucose, lactate, or glutamine[23,44] in a $Na^+$-independent manner. Therefore, our data identifying Cx43 HCs as a previously unknown pathway underlying astroglial glutamine efflux likely accounts for the unknown mechanisms observed in the in vitro study. We indeed found that the specific inhibition of Cx43 HC activity also significantly decreased astroglial glutamine transfer by ~50%. Furthermore, activity-dependent increase in glutamine transfer as well as cognitive performance in vivo, were nearly completely abolished by knocking out Cx43 or by acute inhibition of Cx43 HC activity. Together with our dye uptake experiments showing that Cx43 HC activity is enhanced by activity, it is likely that Cx43 HC-mediated glutamine release may even account for more than 50% of glutamine efflux upon synaptic activity. An obvious advantage of using Cx43 HCs over specialized transporters to mediate glutamine release from astrocytes is reduced energy cost for the cell. The diffusion of molecules through a large pore along a concentration gradient does not consume ATP, and is thus more energetically favorable than release via transporters.

In summary, we generated and utilized a fluorescent probe RhGln to directly observe an activity-dependent transfer of glutamine from astrocyte to hippocampal excitatory synapses and demonstrated a fundamental requirement of the astroglial glutamine supply during physiology. We also showed that this process involves unconventional release pathway, mediated by the activity-dependent opening of Cx43 HCs, and confers physiological functional relevance for excitatory synaptic transmission and recognition memory. Our findings thus reveal an essential function for perisynaptic astrocytes in supplying glutamine as a fuel to synapses and has important implications for future studies on astroglial contributions to sustaining physiological brain functions. While astroglial Cx43 dysfunction has been implicated in neurodegenerative disorders like Alzheimer's disease, glioma, and ischemia[43], our data also provide an astrocyte basis for cognitive disorders. Furthermore, it has been shown that a disrupted excitatory/inhibitory balance is associated with autism spectrum disorders (ASD)[52] and that the disruption of the glutamate–glutamine cycle may trigger the disease onset[53]. Interestingly, astrocyte dysfunction have also been linked to ASD[54], where plasma glutamine level is found to be reduced in patients[55]. Cognitive tasks like memory encoding[56] and consolidation[57] also depend on glutamine and major components of the glutamate–glutamine cycle. Its dysregulation during development has also been shown to have long-lasting impact on spatial memory in adult mice[58]. Our findings showing that exogenous glutamine can rescue defects in synaptic transmission caused by the lack of astroglial supply of glutamine would support the use of glutamine as a supplement to control diseases like ASD, as well as to treat or prevent memory impairment.

## Methods

**Animals**. All procedures on animals were performed according to the guidelines of European Community Council Directives of 01/01/2013 (2010/63/EU) and our local animal care committee (Center for Interdisciplinary Research in Biology in College de France). Experiments were carried out using mice of wild-type C57BL/6j background, mice expressing enhanced green fluorescent protein under the astrocytic promoters glial fibrillary acidic protein (GFAP-eGFP) or aldehyde dehydrogenase 1 family member L1 (Aldh1l1-eGFP), as well as mice with glial conditional deletion of connexin 43 (Cx43) in astrocytes Cx43$^{fl/fl}$:hGFAP-Cre (Cx43−/−). GFAP-eGFP and Cx43−/− mice were provided by F. Kirchhoff (University of Saarland, Germany) and K. Willeke (University of Bonn, Germany), respectively, and were previously characterized[59,60]. Bac Aldh1l1-eGFP mice were obtained from The Jackson Laboratory (GENSAT project; Stock No: 030247) and were characterized[61]. Kir4.1 fl/fl:hGFAP-Cre (Kir4.1−/−) mice with conditional deletion of Kir4.1 in glia were provided by K.D. McCarthy (University of North Carolina, USA)[62,63]. Experimental mice were housed on a 12 h dark/light cycle at 20–21 °C and 45–65% humidity. Mice of both genders and littermates were used at postnatal days (P) 17–30 or 3 months of age. All efforts were made to minimize the number of animals used and their suffering.

**Recombinant adeno-associated virus (rAAV) generation and stereotaxic surgery**. For rAAV in vivo gene transfer, a transgene composed of GFP and Cx43 cDNA separated by a P2A sequence in a single open reading frame was placed under the control of a GFAP-specific promoter in a rAAV shuttle plasmid containing the inverted terminal repeats (ITR) of AAV2 (AAV-GFAP-Cx43-GFP). Pseudotyped serotype 9 rAAV particles were produced by transient co-transfection of HEK-293T cells, as previously described[64]. Viral titers were determined by quantitative PCR amplification of the ITR on DNase-resistant particles and expressed as vector genome per ml (vg/ml). 10 Cx43−/− mice (P15-17) were deeply anesthetized using mixture of ketamine (95 mg/kg; Merial) and xylazine (10 mg/kg; Bayer) in 0.9% NaCl and fitted into a stereotaxic frame (David Kopf Instruments). 0.5 µl of either rAAV2/9-GFAP-Cx43-GFP or rAAV2/9-GFAP-GFP ($1.5 × 10^{12}$ vg/ml) was unilaterally injected into the hippocampus at the rate of 0.1 µl/min using the following coordinates from Bregma: antero-posterior −1.95 mm; medio-lateral −1.5 mm; dorso-ventral −1.38 mm. The injection was performed using 29-gauge blunt-tip needle connected to 2 µl Hamilton syringe and the injection rate was controlled by syringe pump (KD Scientific). After the injection, the needle was left in place for 5 min and then slowly withdrawn.

Following surgery, mice were allowed to recover from anesthesia on a heating pad and monitored for the next 24 h. Mice were sacrificed 13–15 days post-surgery for electrophysiology or immunohistochemistry experiments.

**Antibodies, immunohistochemistry, and immunoblotting**. All antibodies were commercially available and validated by manufacturers. The following primary antibodies were used: polyclonal chicken anti-GFP (1:500, AB13970, Abcam), monoclonal mouse anti-VGlut1 (1:200, 135511, Synaptic Systems), monoclonal mouse anti-Cx43 (1:500, 610-062, BD Biosciences), polyclonal rabbit anti-Cx43 (1:500, 71-2200, Zymed Laboratories), monoclonal mouse anti-GFAP (1:500, G3893, Sigma), and monoclonal mouse anti-GAPDH-peroxidase (1:1000, G9295, Sigma). The following secondary antibodies were used for immunohistochemistry: goat anti-mouse IgG conjugated to Alexa 488, 555 or 647 (1:2000, A11029, A21424 or A21235, Life Technologies), goat anti-rabbit IgG conjugated to Alexa 488 or 555 (1:2000, A11034 or A21429, Life Technologies), and goat anti-chicken IgG conjugated to Alexa 488 (1:2000, A11039, Life Technologies). The following secondary antibodies were used for immunoblotting: goat anti-mouse IgG-HRP (1:2500, sc-2005, Santa-Cruz) and goat anti-rabbit IgG-HRP (1:2500, sc-2004, Santa-Cruz). Immunohistochemistry and immunoblotting were performed as previously described[65,66]. Briefly, acute hippocampal slices (300–400 μm) from 4 +/+ and 8 GFAP-eGFP mice were fixed in 4% paraformaldehyde (PFA) at room temperature for 1 h then blocked for 1 h in phosphate-buffered saline (PBS) containing 1% Triton-X100 and 1% gelatin. Slices were then incubated in appropriate primary antibodies at 4 °C overnight followed by secondary antibodies at room temperature for 2 h the next day. Sections were mounted in Fluoromount-G™ (Thermo Fisher, USA) for image acquisition. Western immunoblotting was carried out on fresh hippocampal tissue or synaptosomal fractions isolated from mice (4 +/+ and 4 Cx43−/− mice). Equal amounts of protein was then separated by electrophoresis in a 10% polyacrylamide gel. This was followed by transfer of proteins onto nitrocellulose membranes which were then saturated with 5% fat-free dried milk in triphosphate buffer solution and incubated overnight at 4 °C with appropriate primary antibodies. On the next day, membranes were treated with peroxidase-conjugated secondary antibodies for 2 h at room temperature and revealed using a chemiluminescence detection kit (ECL, GE Healthcare) and visualized using LAS 4000 imaging system and ImageQuant LAS 4000 software (Fujifilm, USA).

**Synthesis, structural confirmation, and characterization of rhodamine-tagged glutamine**. The red fluorescent molecule rhodamine provides high molar absorptivity, fluorescent quantum yield, and photostability and was therefore selected as the fluorescent tag of choice for glutamine. In addition, having a probe in the red spectrum allows it to be used in transgenic mice expressing green fluorescent protein under specific promoters. For this study, rhodamine-tagged-glutamine (RhGln) molecule was synthesized in a 5-step chemical reaction detailed in Supplementary Fig. 1. For structural confirmation of the intermediate compounds 3 and 4 and final product RhGln, proton, and carbon-13 nuclear magnetic resonance (1H and 13C NMR) spectrometry as well as mass spectrometry were used (Supplementary Fig. 2). The 1H and 13C spectra were collected on a JEOL-FT NMR 400 MHz (100 MHz for 13 C NMR) spectrophotometer equipped with Delta software (Jeol) using CDCl3 or CD3OD as a solvent and tetramethylsilane as an internal standard. Mass spectrometry analyses were performed by Small Molecule Mass Spectrometry platform of ICSN (Centre de Recherche de Gif-sur-Yvette, Universite Paris-Saclay, France). All chemicals and reagents were purchased from Sigma-Aldrich and used without any further purification. 4-(2-phthalimidoethoxy) rhodamine (compound 3): a solution of 1 (0.31 g, 1.06 mmol) in propionic acid (5 mL) was stirred at room temperature followed by the addition of 8-hydroxyjulolidine (0.40 g; 2.13 mmol) and PTSA (p-toluenesulfonic acid; catalytic amount). The resulting reaction mixture was protected from light and stirred at room temperature for 12 h. Saturated sodium acetate solution was added to the reaction mixture until pH reached 7–8, resulting in the precipitation of intermediate compound 2. The precipitate was filtered, washed with water, dried, and used immediately for the next step. The crude product 2 (0.60 g; 0.91 mmol) was dissolved in dry CH2Cl2 (20 mL) followed by the addition of chloranil (0.33 g; 1.37 mmol) and allowed to stir for 12 h at room temperature. The solvent was removed under reduced pressure and the residue was subjected to column chromatography (CH2Cl2/MeOH; 9.5/0.5). Compound 3 was obtained as a purple solid (79% over 2 steps). 1H NMR (CDCl3, 400 MHz): $\delta = 7.87$–7.89 (m, 2 H, Ar-H), 7.74–7.76 (m, 2 H, Ar-H), 7.21 (d, $J = 8$ Hz, 2 H, Ar-H), 7.08 (d, $J = 8$ Hz, 2 H, Ar-H), 6.82 (s, 2 H, Ar-H), 4.35 (t, $J = 8$ Hz, 2 H, CH2), 4.18 (t, $J = 8$ Hz, 2 H, CH2), 3.51–3.58 (m, 8 H, CH2), 3.02 (t, $J = 8$ Hz, 4 H, CH2), 2.69 (t, $J = 8$ Hz, 4 H, CH2), 2.10 (t, $J = 4$ Hz, 4 H, CH2), and 1.97 (t, $J = 4$ Hz, 4 H, CH2) ppm. 13C NMR (CDCl3, 100 MHz): $\delta = 168.31$, 159.55, 154.72, 154.66, 152.31, 151.11, 134.35, 132.04, 131.09, 126.81, 125.17, 123.57, 114.96, 113.00, 105.50, 65.01, 51.04, 50.59, 37.28, 27.78, 20.77, 20.02, and 19.84 ppm. HRMS (TOF MS ES+): found: [M]+ = 636.2865; Calc. for $C_{41}H_{38}N_3O_4{}^+$: [M]+ = 636.2857. 4-(2-aminoethoxy) rhodamine (compound 4): to a solution of compound 3 (0.12 g, 0.19 mmol) in absolute ethanol (5 mL) was added hydrazine monohydrate (0.09 g, 1.95 mmol). The resulting mixture was refluxed for 4 h. The solvent was removed under reduced pressure and 20 ml of CH2Cl2 was added to the mixture. The organic layer was washed with 10 ml of NaOH solution ($10^{-4}$ mol L$^{-1}$) and was dried with MgSO4 to provide 70 mg of free amine 4 (82%), which was used without

any further purification. 1H NMR (CD3OD, 400 MHz): $\delta = 7.33$ (d, $J = 8$ Hz, 2 H, Ar-H), 7.21 (d, $J = 8$ Hz, 2 H, Ar-H), 6.94 (s, 2 H, Ar-H), 4.21 (t, $J = 4$ Hz, 2 H, CH2), 3.70 (t, $J = 4$ Hz, 2 H, CH2), 3.50–3.56 (m, 8 H, CH2), 3.06 (t, $J = 8$ Hz, 4 H, CH2), 2.72 (t, $J = 8$ Hz, 4 H, CH2), 2.08–2.11 (m, 4 H, CH2) and 1.95–1.96 (m, 4 H, CH2) ppm. 13C NMR (CD3OD, 100 MHz): $\delta = 160.17$, 154.90, 152.31, 151.02, 131.08, 126.46, 125.02, 123.79, 114.53, 112.66, 105.27, 50.48, 49.98, 40.16, 29.42, 27.29, 20.48, 19.63 and 19.53 ppm. HRMS (TOF MS ES+): found: [M] + = 506.2811; Calc. for $C_{33}H_{36}N_3O_2{}^+$: [M]+ = 506.2802. 4-(2-(N-glutamino) ethoxy) rhodamine (RhGln): compound 4 (0.10 g, 0.21 mmol), Boc-Glu-OtBu (0.06 g, 0.21 mmol), EDCI.HCl (0.04 g, 0.23 mmol) and DMAP (0.002 g, 0.02 mmol) were dissolved in dry DMF (5 mL) under argon atmosphere. The mixture was stirred at room temperature for 12 h followed by solvent removal under reduced pressure. The residue was dissolved in CH2Cl2 (50 mL) and washed with water (100 mL). The organic layer was dried over Na2SO4 and concentrated under vacuum. The residue was then dissolved in a mixture of CH2Cl2-TFA (10 mL; 8:2 v/v) and the resulting mixture was allowed to stir at room temperature for 1 h. The solvent was removed under reduced pressure and the remaining solid was subjected to column chromatography (CH2Cl2/MeOH; 9/1). RhGln was obtained as a purple solid (60% over 2 steps). 1H NMR (CD3OD, 400 MHz): $\delta = 7.34$ (d, $J = 8$ Hz, 2 H, Ar-H), 7.21 (d, $J = 8$ Hz, 2 H, Ar-H), 6.94 (s, 2 H, Ar-H), 4.19 (t, $J = 8$ Hz, 2 H, CH2), 3.75–3.78 (m, 1 H, CH), 3.67 (t, $J = 4$ Hz, 2 H, CH2), 3.51–3.57 (m, 8H, CH2), 3.07 (t, $J = 8$ Hz, 4 H, CH2), 2.88–2.91 (m, 2 H, CH2), 2.73 (t, $J = 8$ Hz, 4 H, CH2), 2.52 (t, $J = 4$ Hz, 2 H, CH2), 2.09–2.17 (m, 6 H, NH2 and CH2) and 1.97–1.98 (m, 4 H, CH2) ppm. 13C NMR (CD3OD, 100 MHz): $\delta = 173.84$, 164.99, 160.15, 154.98, 152.35, 151.05, 131.05, 126.47, 123.81, 114.49, 112.68, 105.28, 66.45, 53.79, 50.49, 49.99 42.20, 38.78, 31.47, 27.30, 26.43, 20.49, 19.63, and 19.54 ppm. HRMS (TOF MS ES+): found: [M]+ = 635.3220; Calc. for $C_{38}H_{43}N_4O_5{}^+$: [M]+ = 635.3228.

Density functional theory (DFT) calculations were performed using the m062x method and the 6–31G+ (d) base with IEFPCM(water) for the solvent. RhGln is functionalized at the amide function of glutamine by a rhodamine derivative. The chemical link between rhodamine and glutamine group is an alkyl chain. Although these compounds have different solvent-accessible volumes (169.56 Å$^3$ for Gln and 737.64 Å$^3$ for RhGln), the choice of rhodamine as a fluorophore was governed by its interesting photophysical properties (absorption wavelength >550 nm, high fluorescence quantum yield, possibility of high-resolution microscopy). The fluorescence properties of RhGln were characterized using both steady state and time-resolved measurements. For steady-state measurements, samples were prepared in 1 cm$^2$ quartz optical cell using the appropriate solvent (i.e., chloroform, ethanol, or intracellular solution). UV/Vis absorption spectra were recorded on a Varian Cary 5000 spectrophotometer and corrected emission spectra were collected on a Jobin-Yvon SPEX Fluoromax-4 spectrofluorometer using samples with optical densities below 0.1 at the excitation wavelength, as well as at all emission wavelengths. The RhGln exhibits a strong $S_0 \rightarrow S_1$ transition in the green-to-red region of the visible spectra with a corresponding molar absorptivity of $1.2 \times 10^5$ M$^{-1}$.cm$^{-1}$ at 580 nm. The fluorescence quantum yields were determined using Rhodamine 101 in methanol as standard[67]. For time-resolved measurements, fluorescence emission decays were obtained using a time-correlated single-photon counting (TCSPC) apparatus. Samples were prepared in 1 cm$^2$ quartz optical cell using the appropriate solvent (i.e., chloroform, ethanol, or intracellular solution). Excitation was achieved using a Spectra-Physics setup composed of a Ti:Sa Tsunami laser (FWHM = 100 fs) pumped by a frequency-doubled Millennia Nd:YVO4 laser. Light pulses were selected by optoacoustic crystals at a repetition rate of 4 MHz and a frequency doubling crystal is used to reach the desired excitation wavelength of 495 nm (FWHM = 200 fs; 200 μW). Emitted photons were detected, through a monochromator centered at 600 nm, using a Hamamatsu MCP R3809U photomultiplier connected to a constant-fraction discriminator. The time-to-amplitude converter was purchased from Tennelec. The fluorescence data were analyzed by a nonlinear least-squares method on Globals software developed at the Laboratory of Fluorescence Dynamics at the University of Illinois, Urbana-Champaign. Satisfactory fits of time-resolved fluorescence measurements in intracellular solution were obtained by considering a bi-exponential model, leading to corresponding lifetimes of: $\tau_1 = 4.2$ ns ($a_1 = 97\%$) and $\tau_2 = 0.9$ ns ($a_2 = 3\%$). In contrast, single exponential behavior, with nearly identical lifetime ($\tau = 3.5$ ns), was observed in non-polar organic solvent such as chloroform. Such effect could derive from the presence of different conformers in intracellular solution, where efficient photo-induced electron transfer could occur between the dye and the glutamine moiety.

**Acute hippocampal slice preparation**. Acute transverse hippocampal slices (300–400 μm) were prepared as previously described[68]. Briefly, mice were sacrificed by cervical dislocation and decapitation. The hippocampi were immediately isolated and sectioned at 4 °C using a vibratome (Leica) in an artificial cerebrospinal fluid (ACSF) containing (in mM): 119 NaCl, 2.5 KCl, 2.5 CaCl2, 1.3 MgSO4, 1 NaH2PO4, 26.2 NaHCO3 and 11 glucose (pH = 7.4). Slices were maintained at room temperature in a storage chamber containing ACSF saturated with 95% O2 and 5% CO2 for at least 1 h prior to experiments.

**Electrophysiology and dye-filling of astrocytes in acute hippocampal slices**. Acute hippocampal slices were transferred to a submerged recording chamber mounted on an Olympus BX51WI microscope and were perfused with ACSF at a rate of 2 ml/min at room temperature. Except when indicated, all

electrophysiological recordings were performed in the presence of picrotoxin (100 μM) and CPP (10 μM; Tocris Bioscience), and a cut was made between CA3 and CA1 regions to prevent the propagation of epileptiform activity. Neuronal field recordings were performed as previously described[65]. Field excitatory postsynaptic potentials (fEPSPs) were evoked by Schaffer collaterals stimulation (10–20 μA, 0.1 ms) through an ACSF-filled glass pipette located at a distance of 200 μm from the recording area. Basal activity was measured at 0.1 Hz, whereas enhanced stimulated activity was recorded at 10 Hz for 30 s once or repeated every 3 min for 20 min. For patch-clamp recordings and dye-filling experiments, CA1 astrocytes were visually identified and filled under whole-cell configuration via glass pipettes (5–8 MΩ) containing an intracellular solution (in mM): 105 K-gluconate, 30 KCl, 10 HEPES, 10 phosphocreatine, 4 ATP-Mg, 0.3 GTP-Tris, 0.3 EGTA (pH 7.4, 280 mOsm/l). Stratum radiatum astrocytes were verified by their low input resistance (~20 MΩ), high resting potentials (~−80 mV), and linear passive input/output relationship under voltage-clamp configuration. 0.8 mM of either RhGln or a rhodamine only dye (Rh) was also included for dye-filling experiments and astrocytes were dialyzed for 20 min. In some experiments, pharmacological substances were included in the perfusion ACSF before and during the experiments: a Cx43 hemichannel blocker peptide and its scramble version (Gap26/Gap26 scramble; 100 μM) and a blocker of neuronal glutamine transporter, α-(Methylamino) isobutyric acid (MeAIB, 20 mM) with 5 min pre-incubation; a gap-junction blocker carbenoxelone (CBX, 200 μM) with 10 min pre-incubation; glutamine (4 mM) with 1–4 h pre-incubation in storage chamber). After dye filling, patch pipettes were carefully removed and slices were fixed for 1 h in 4% PFA at room temperature and preserved for immunohistochemistry or directly mounted for image acquisition. Recordings were acquired with Axopatch-1D amplifiers (Molecular Devices, USA), digitized at 10 kHz, filtered at 2 kHz, stored, and analyzed on a computer using pClamp9 and Clampfit10 software (Molecular Devices). fEPSP slopes were normalized to baseline and plotted over 30 s during stimulation. Stimulation artifacts were removed in representative traces for illustration. For electrophysiological recordings, 30 +/+ and 14 Cx43−/− mice were used. For dye filling experiments, 27 +/+, 8 GFAP-eGFP, and 4 Aldh1l1-eGFP mice were used.

**Ethidium bromide uptake assay and RhGln bulk loading experiments**. Acute hippocampal slices (350 μm) were transferred to a submerged recording chamber and incubated in HEPES buffer containing (in mM): 140 NaCl, 5.5 KCl, 1 MgCl$_2$, 1.8 CaCl$_2$, 10 HEPES, 10 C$_6$H$_{12}$O$_6$ (pH = 7.4). Ethidium bromide (EtBr; 314 Da, 4 μM) or RhGln (0.05 mM) was then added extracellularly to the recording chamber. After a 10 min pre-incubation, some slices were electrically stimulated (10 Hz for 30 s every 3 min) at Schaffer collaterals for 20 min via a glass pipette as described for electrophysiological experiments. To investigate the contribution of Cx43 hemichannels to EtBr uptake, slices were pre-treated 5 min before and during EtBr incubation with Gap26 or Gap26 scramble (100 μM). Slices were then rinsed for 15 min in fresh HEPES buffer, fixed for 1 h in 4% PFA at room temperature, and preserved for immunohistochemistry or directly mounted for image acquisition. For EtBr uptake experiments, 30 +/+ and 6 Kir4.1−/− mice were used. For RhGln bulk loading experiments, 4 GFAP-eGFP and 4 Aldh1l1-eGFP mice were used.

**Isolation of synaptosomal fraction**. Crude synaptosomal fractions from hippocampus were isolated for immunoblotting. First, freshly dissected whole hippocampi or acute hippocampal slices were mechanically homogenized using a Potter-Elvehjem homogenizer in ice-cold homogenization buffer (0.32 M sucrose, 10 mM HEPES, 2 mM EDTA, protease inhibitors cocktail; 400 μl/hippocampus). Homogenate was then centrifuged at 900 × g for 15 min at 4 °C. The supernatant (S1) was centrifuged at 16,000 × g for 15 min at 4 °C. The pellet was then washed and resuspended in fresh ice-cold homogenization buffer (300 μl/hippocampus) and centrifuged again at 16,000 × g for 15 min at 4 °C. The resulting pellet containing synaptosomes (P2) was resuspended in HEPES lysis buffer (50 mM HEPES, 2 mM EDTA, protease inhibitors cocktail; 150 μl/hippocampus). Samples were briefly sonicated to ensure membrane lysis and used for immunoblots. For normalization, exact protein concentration of each sample was determined using Ionic Detergent Compatibility Reagent (22663, Thermofisher, France). The purity and composition of our hippocampal synaptosomal fractions have been previously verified and reported[69,70]. We showed that compared to total hippocampal fractions, synaptosomes are enriched in pre- and postsynaptic markers like Synapsin I and PSD95, respectively[69], as well as the perisynaptic astroglial protein ezrin[70]. They are also immunopositive for VGlut1 and Homer1[70].

**Immunogold labeling and serial section electron microscopy**. Hippocampal tissue preparation for ultrastructural analysis using serial section electron microscopy was previously described[65]. Briefly, mice were anaesthetized with Narko-dorm® (60 mg/kg body weight) and then transcardially perfused with physiological saline followed by an ice-cold 0.1 M phosphate-buffered (pH 7.4) fixative containing 4% PFA and 0.1% glutaraldehyde (Polyscience Europe GmbH, Eppelheim, Germany) for 15 min. To immunolabel Cx43 with gold particles, 100 μm vibratome sections containing the hippocampus were made and treated with a monoclonal mouse anti-Cx43 antibody (1:250, Chemicon Europe, Hampshire, UK) overnight at 4 °C followed by a 15 nm gold-conjugated anti-mouse secondary antibody (1:40, British Biocell, Cardiff, UK) on the next day for 2 h. 10–30 serial ultrathin sections

(55 ± 5 nm) were cut using a Leica Ultracut S ultramicrotome (Leica Microsystems, Vienna, Austria) and collected on Formvar-coated slot copper grids. Prior to electron microscopic examination, sections were stained with 5% aqueous uranyl acetate for 20 min and lead citrate for 5 min (Reynolds 1963). Ultrathin sections were inspected and photographed with a Zeiss Libra 120 electron microscope (Fa. Zeiss, Oberkochen, Germany) equipped with a Proscan 2K digital camera and the SIS Analysis software (Olympus Soft Imaging System, Münster, Germany). Individual digital images were further process for final illustrations using the Adobe Photoshop™ and Illustrator™ software packages. For the analysis of average distances between gold particles and the edge of the nearest active zone, 75 synaptic complexes from 8 ultrathin sections were used. All Cx43 positive gold particles identified on the images were counted using ImageJ software (National Institutes of Health, USA). The astrocytic profiles that are near or in close proximity can be at the shortest distance membrane to membrane (<20 nm) with the synaptic profiles composed of the synaptic bouton and the postsynaptic spine.

**Confocal image acquisition and analysis**. Fixed and mounted acute brain slices were examined using a confocal laser-scanning microscope (TCS SP5, Leica) and image acquisition software LAS X (Leica). To avoid high surface background, acquisition was always made 10–30 μm below slice surface. Fluorescence images were acquired in sequential mode with different excitation lasers (Argon 488 nm, DPSS 561 nm, and HeNe 633 nm). Three different objectives were used: ×20 objective (NA = 0.7) was used to capture single-plane image overviews of the entire CA1 region; ×40 objective (NA = 1.25) was used to obtain z-stacks of 1-μm intervals to focus on selected areas in this region; ×63 objective (NA = 1.4–0.6) at 2.5 x zoom was used to obtain z-stacks of 0.2-μm intervals to closely examine morphology and subcellular compartments. Confocal image analysis was performed using ImageJ software v1.52j. The size of RhGln-labeled network was determined by counting the number of RhGln-labeled cells in individual experiments and averaging across experiments for each condition. To aid visualization for illustration purposes, RhGln signal in a network of cells were subjected to local contrast normalization using an *Integral Image Filters* plugin and thresholded by intensity. In Fig. 2a, sample images are presented as maximum z-projection. To quantify the extent of RhGln-labeling from the soma of the dye-filled astrocyte, high-resolution image stacks (obtained using ×63 objective) containing single patched astrocytes were analyzed. A mask of each image of the stacks was generated by thresholding first by intensity and then by size (0.5–2 μm$^2$) using *Subtract background*, *Threshold*, and *Analyze Particles* plugins. The same thresholding range was used for images from different experimental conditions to allow direct comparison. The maximum projection of the masks for each astrocyte was then analyzed using *Sholl Analysis* plugin. Results were given as the number of intersections found on each concentric circle from the center of the cell soma at 1 μm-radius interval. The fitted curve of the number of intersections over distance for each cell was then extracted and averaged. In addition, total punctate area for each cell was also measured from thresholded image masks. For co-localization analysis, fluorescence images from each channel were first separately deconvolved using Huygens software (Scientific Volume Imaging B.V., The Netherlands). Image masks based on intensity thresholding were then generated for each channel and merged to find co-localized areas representing presynaptic structures containing RhGln. For the co-localization between RhGln and VGlut1, the percentage of co-localization was measured as the co-localized areas normalized to the total RhGln-positive area of the area of interest. For the determination of the % of perisynaptic Cx43, total Cx43 was first determined as the co-localized area between GFAP-eGFP and Cx43. Then, the perisynaptic population of Cx43 was calculated based on the area which is further co-localized with VGlut1 and then normalized to total Cx43. Analysis of EtBr uptake was performed on stratum radiatum CA1 GFAP-positive astrocytes. Fluorescence intensity was measured as integrated density in arbitrary units and corrected to background fluorescence. For statistical quantifications, 60 CA1 astrocytes were analyzed per acute slice, and data were collected from 4 Aldh1l1-eGFP mice. For EtBr incubation experiments without electrical stimulation, 20 CA1 astrocytes were analyzed per individual field of view. For the analysis of astrocytic morphology, image stacks containing single astrocytes immunostained for GFAP were processed by *Sholl Analysis*. After intensity thresholding, images were inverted and the number of intersections of astrocytic processes on each concentric circle from the center of the cell soma at 5 μm radius interval was determined. In addition, astrocytic soma and domain sizes were determined by delineating the visible cell body area of Aldh1l1-eGFP signal and entire GFAP-positive 2D coverage, respectively.

**Super-resolution stimulated emission depletion (STED) imaging in brain slices**. STED imaging was performed to acquire high-resolution images of RhGln-labeled astrocytes in acute brain slices at the level of fine astroglial processes and synaptic structures after immunostaining. Our custom-made upright STED microscope (Abberior Instruments) based on a Scientifica microscope body (Slice Scope, Scientifica) was equipped with an Olympus 100X/1.4NA ULSAPO objective lens. It comprises of a scanner design featuring four mirrors (Quad Scanner, Abberior Instruments) with three excitation lasers at 488 nm, 561 nm, and 640 nm (Abberior Instruments, pulsed at 40/80 MHz). Two additional lasers at 595 nm (MPB-C, cw), and 775 nm (MPB-C, pulsed at 40/80 MHz) were used to generate STED beams. The conventional laser excitation and STED laser beams were superimposed using a

beam-splitter (HC BS R785 lambda/10 PV flat, AHF Analysetechnik). Common excitation power with pulsed excitation ranging from 10 to 20 µW with STED power intensities of up to 200 mW in the focal plane were used. Super-resolution images were acquired with pixel size of 30 nm, 15–25 µm dwell time with three times averaging using acquisition software Inspector 16.4 (Abberior Instruments).

**Novel object recognition test**. Novel object recognition (NOR) was assessed in 3-month-old wild-type male mice using a NOR apparatus consisting of a dark gray polypropylene box (40 × 30 × 23 cm; length, width, height). A glass rectangle (5 × 5 × 5 cm) and a ceramic bowl (8 cm in diameter, 5 cm in height) were used for object recognition; they were too heavy to be displaced by the animal and located at two corners of the apparatus. The mice did not show any preference for either of the two objects. For the implantation of cannulas, mice were first deeply anesthetized with ketamine (95 mg/kg) and xylazine (10 mg/kg) via intraperitoneal injections and stereotaxically implanted with a 26-gauge bilateral guide cannula (2 mm long below pedestal, Plastic One, Roanoke, VA, USA) into the hippocampus using the following coordinates from Bregma: antero-posterior −2 mm; medio-lateral ±1.5 mm; dorso-ventral −1 mm from cortex. This was then held in place with dental cement. After surgery, animals were re-housed in home cage to recover at least for 7 days with close monitoring. Behavioral testing was conducted during the light phase of the 12 h light/dark cycle under dim illumination (50lux) and recorded using LifeCam Studio camera and software (Microsoft). Prior to the first NOR test, during three consecutive days animals were handled 1 min per day. On the day preceding each NOR test, animals were allowed to freely explore the empty (without objects) apparatus for 10 min for habituation. The NOR test consisted of 2 trials (T1 and T2) separated by an inter-trial time interval (ITI). To assess the effect of acutely blocking Cx43 hemichannels and rescue by glutamine on NOR performance, mice were administered a bilateral intra-hippocampal injection (1 µL per side; 1 µL/min) of either Gap26 (1 mM), Gap26 scramble (1 mM) or Gap26 (1 mM) + glutamine (200 mM) 30 min before T1 ($n = 10$ per group). This was carried out using a precision pump with a 33-gauge bilateral cannula (Plastics One) extending 0.5 mm below the end of the guide cannula to target hippocampus CA1 stratum radiatum (from Bregma: antero-posterior −2 mm; medio-lateral ±1.5 mm; dorso-ventral −1.5 mm from cortex). On T1 (acquisition trial), subjects were placed in the apparatus containing two identical objects (F1 and F2) for 10 min before being returned to their home cage for an ITI of 24 h. Then, they were placed back in the NOR apparatus containing a familiar (F) and a novel object (N) for 5 min (T2, restitution trial). The type (familiar or novel) and the position (left or right) of the two objects were counterbalanced and randomized within each experimental group during T2. Between each trial, the NOR apparatus was cleaned with water and the objects with 70% ethanol. Exploration was defined as the animal directing the nose within 0.5 cm of the object while looking at, sniffing or touching it, excluding accidental contacts with it (backing into, standing on the object, ...). The raw exploration data were normalized to the total objects exploration time.

**Statistical analysis**. Data was stored and analyzed using Microsoft Excel (Microsoft). Statistical analysis were performed using GraphPad Prism software v7 and v8 (USA). All data are expressed as mean ± standard error of the mean (SEM), and $n$ represents the number of independent experiments, unless otherwise stated. Statistical significance was determined by one-sampled $t$-test, two-tailed unpaired Student's $t$-tests, Mann–Whitney test, one-way ANOVA with Bonferroni post hoc tests, Kruskal–Wallis test with Dunn's post hoc test, or two-way ANOVA. The test for normality was performed using Shapiro–Wik normality test. Individual data points are plotted in all bar graphs with p-values given in figure legends.

**Drugs**. Gap26 (VCYDKSFPISHVR), and Gap26 scramble (PSFDSRHCIVKYV) were synthesized by Thermo Fisher Scientific (purity, 95%), and all other products were from Sigma unless otherwise stated. LY341195 and CPP were purchased from Tocris Biosciences.

**Reporting summary**. Further information on research design is available in the Nature Research Reporting Summary linked to this article.

## Data availability
We confirm that all relevant data are included in the paper and/or its supplementary information files. Source data are provided with this paper.

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

## Acknowledgements

We thank D. Mazaud and. J. Cazères for technical assistance. This work was supported by grants from the European Research Council (Consolidator grant #683154) and European Union's Horizon 2020 research and innovation program (Marie Sklodowska-Curie Innovative Training Networks, grant #722053, EU-GliaPhD) to N.R. and from FP7-PEOPLE Marie Curie Intra-European Fellowship for career development (grant #622289) to G.C.

## Author contributions

Conceptualization: N.R. and G.C.; data curation: G.C. and N.R.; formal analysis: G.C., J.M., G.D, O.C., J.V., and D.M.; funding acquisition: N.R. and G.C.; investigation: G.C., D.B., J.M., G.D., J.V., A.R., O.C., P.E., and D.M.; methodology: G.C., D.B., N.K., and J.L.; resources: N.K., C.M., A.B., and I.L.; supervision: N.R., I.L., and J.L.; project administration: N.R.; validation: G.C., D.B., and N.R.; visualization: G.C., C.M., G.D., and N.R.; writing—original draft: G.C. and N.R.; writing—review & editing: G.C. and N.R.; J.M. and G.D. equally contributed to this study.

## Competing interests

The authors declare no competing interests.
