## [Peer Review File · Nature Communications]

Physiological synaptic activity and recognition memory require astroglial glutamineREVIEWER COMMENTS

Reviewer #1 (Remarks to the Author):

The article by Cheung et al reports on the use of a novel fluorescent rhodamine-tagged glutamine molecule (RhGln) to visualize the dynamics of glutamine at the cellular level in the brain. Using this tool the authors demonstrate that physiological synaptic activity results in a transfer of glutamine from astrocytes to neurons mediated by Connexin 43 (Cx43) hemichannels (HC). They also show that such a dynamic transfer is required for object recognition memory. The study has been carefully conducted with techniques that are well described and appropriate. While there are only a few points that would need clarification, the main issue of this work is that it provides mainly incremental information on a metabolic and signaling pathway that has essentially been described over 50 years ago and is already in neuroscience textbooks. Granted that some refinements are provided with contemporary sophisticated techniques, but the article does not represent a major conceptual advance. The identification of the signaling mechanism(s) for the activity dependent function of Cx43 HC would have been a novel and original contribution to the topic.

Specific points :

Page 5, line 20 : "fine astroglial processes are often found near synaptic structures..." . The adverb often, is not ideal; mainly would be more appropriate.

Page 6, lines 22-23: Some documentation about the purity and cellular composition of the synaptosomal fractions should be provided.

Page 6, lines 10-11 : what is the signaling mechanism that triggers the HC function of Cx43 to mediate glutamine release?

Page 8, lines 21-22 : addition of glutamine at 4 mM rescues genetically- or pharmacologically-induced impairment of Cx43 function. It would be interesting to know the kinetics of the glutamine transporter in neurons and how 4 mM relates to it.

Page 9, line 2-3: In the introduction the authors indicate that without a glutamine recycling process glutamatergic transmission would be exhausted within one minute of basal synaptic activity. The fluorescent probe used has been chemically designed to prevent the conversion of glutamine to glutamate. The question then is how much the probe interferes with normal neuronal glutamate replenishment as this process cannot be provided by the stable fluorescent probe. Some quantitative data should be provided to address this question.

Page 14, lines 15-16: It would be important to have a quantitative assessment of what % of total Cx43 are present on astrocytic processes around synaptic profiles.

Page 16, lines 17-18: It would be useful to have some examples of diseases in which Cx43 function is altered.

Reviewer #2 (Remarks to the Author):

This is an elegant and thrilling study based on the development and smart application of a novel fluorescent-labeled glutamine analog. The authors dubbed the rhodamine-tagged glutamine molecule, RhGln, and convincingly demonstrated the usability of this tool. Particularly, RhGln was shown to enable the high-resolution visualization of its transfer from astrocytes to presynaptic structures in vitro under basal conditions and upon synaptic stimulation. By several additional smart and well-designed experiments, the authors eventually unveiled the molecular mechanism and physiological role of this directed glutamine transport.

I have some minor comments/suggestions to be considered by the authors:

- 1.) The authors might also show if the diffusion of RhGln toward neighboring cells is RhGln specific. Does the unconjugated rhodamine dye also diffuse in the same manner (extend and kinetics) in this experimental setting?
- 2.) The authors might better elaborate and discuss how the conjugation of the relatively large rhodamine molecule impacts the transport and metabolism of the amino acid. What might be important differences in the biochemistry between Gln and RhGln? What needs to be considered

when interpreting respective results based on RhGln? What are the real limitations of using RhGln? 3.) The authors might additionally discuss the potential clinical/medical implications of their findings that show the restoring effects of supplementing synapsis with exogenous glutamine.

Reviewer #3 (Remarks to the Author):

This paper by Cheung et al. studies glutamine dynamics on astrocyte networks and glutamine transfer from astrocytes to neurons. For this matter, the authors have developed a novel tool that allows them to visualize glutamine and trace it (a fluorescent form of glutamine that they call RhGln). They perform imaging studies combined with electrophysiology and behavior and found that glutamine transfer from astrocytes to neurons is necessary for a correct vesicle replenishment to maintain the presynaptic glutamate release. I believe this paper represents an advance in the field, and sheds light in the glutamine-glutamate cycle. In general, I consider that the experimental approach is appropriate and that the main conclusions are in accordance with the data presented, however, I have some comments and suggestions that I hope will help to strengthen the manuscript:

The authors claim that RhGln-labeling is not due to extracellular uptake as bulk loading of slices with RhGln did not label astrocytes. However, the chosen image in Suppl figure 2 to illustrate this is not very convincing. The GFAP-GFP labeling is very sparse and it's difficult to know if the red labeled cells in the stratum radiatum are astrocytes or interneurons. In this point, I think it is also important to point out that neurons do uptake RhGln from the extracellular space, as we can see in the images.

In experiments in Fig2d-e the authors found that synaptic stimulation decreases the glutamine intercellular transfer. But it is possible that this effect is not specific for RhGln, in fact, is it possible that this effect is due to the closing of the hemichannels during the stimulation? A control experiment loading rhodamine would be necessary.

For Fig 4f the manuscript would benefit if the authors provide an explanation for the use of ethidium bromide. For this experiment the authors claim that "this uptake was mediated by Cx43 HCs, as it was abolished by the Gap26 peptide" but it is not abolished, it is reduced compared to control conditions. In my opinion the fact that Gap26 peptide reduces uptake below control conditions is important as it indicates that there is a basal glutamine transfer from the astrocytes to the synapse.

They have found that glutamine transfer is necessary for replenishment of the presynaptic pool of glutamate, and they have done this under a specific synaptic stimulation, but they have not tested whether there is a change in basal synaptic transmission. Therefore, sentences like "Importantly, the same exogenous glutamine treatment in +/+ slices had no effect on excitatory transmission (Fig. 7h)" or "we identified astroglial Cx43 HCs as the key mediators of this process, essential for sustaining physiological excitatory synaptic transmission as well as recognition memory" are somehow misleading. One possibility is to perform experiments where they analyze amplitude and frequency of spontaneous or miniature EPSCs, otherwise I feel the authors should down tone the manuscript regarding the role of glutamine on excitatory transmission.

Response to Reviewers

We thank all reviewers for their positive feedback on our manuscript. We also greatly appreciate their constructive comments and insightful suggestions to improve our study. We have now revised our manuscript with a series of additional experiments to address every point raised. Below is our point-by-point response to each reviewer.

Point-by-point response to Reviewer #1

Reviewer #1 (Remarks to the Author):

The article by Cheung et al reports on the use of a novel fluorescent rhodamine-tagged glutamine molecule (RhGln) to visualize the dynamics of glutamine at the cellular level in the brain. Using this tool the authors demonstrate that physiological synaptic activity results in a transfer of glutamine from astrocytes to neurons mediated by Connexin 43 (Cx43) hemichannels (HC). They also show that such a dynamic transfer is required for object recognition memory. The study has been carefully conducted with techniques that are well described and appropriate.

We thank the reviewer for acknowledging the appropriateness of our experimental approach and quality of our work.

While there are only a few points that would need clarification, the main issue of this work is that it provides mainly incremental information on a metabolic and signaling pathway that has essentially been described over 50 years ago and is already in neuroscience textbooks. Granted that some refinements are provided with contemporary sophisticated techniques, but the article does not represent a major conceptual advance. The identification of the signaling mechanism(s) for the activity dependent function of Cx43 HC would have been a novel and original contribution to the topic.

Although the glutamate-glutamine cycle has been proposed years ago, and well-admitted in pathological conditions, the presence of an astroglial glutamine supply as well as its functional relevance in physiological conditions and *in vivo* in the healthy brain remained controversial, primarily due to the lack of tools that can directly track glutamine transfer and to the use of unspecific pharmacological agents. Our study, by developing a novel glutamine fluorescent probe, here reveals: 1) direct visual evidence for an activity-dependent astrocyte-neuron transfer of glutamine under physiological conditions, 2) functional relevance of this transfer in physiological synaptic transmission and memory, 3) an entirely novel molecular and now signaling mechanism (see below) underlying this astroglial supply in physiological conditions. We indeed uniquely demonstrate the involvement of Cx43HC in activity-dependent astroglial glutamine supply, which has never been shown previously to be linked to the glutamate-glutamine cycle.

In response to the concern of the reviewer that our article in its initial form did not represent a major conceptual advance, and that identification of the signaling mechanisms for the activity-dependent opening of Cx43 HC would contribute to the novelty and originality of our work, we have now performed a series of experiments to identify this mechanism (please see details below). With the additional elucidation of the signaling mechanism triggering Cx43HC opening to allow glutamine

release in an activity-dependent manner, as suggested by the reviewer, we believe that our discoveries as reported in our revised manuscript provide significant conceptual advancement to the understanding of the role of astrocytes in sustaining physiological transmission.

Specific points:

Page 5, line 20: “fine astroglial processes are often found near synaptic structures...” . The adverb often, is not ideal; mainly would be more appropriate.

This sentence has been modified as suggested.

See Page 6, line 133 in *Results* section:

“... fine astroglial processes are mainly found near synaptic structures ...”

Page 6, lines 22-23: Some documentation about the purity and cellular composition of the synaptosomal fractions should be provided.

The purity and cellular composition of the synaptosomal fractions is now provided.

See Page 30, lines 670-674 in *Methods* section:

“The purity and composition of our hippocampal synaptosomal fractions have been previously verified and reported^{69,70}. We showed that compared to total hippocampal fractions, synaptosomes are enriched in pre- and postsynaptic markers like Synapsin I and PSD95, respectively⁶⁹, as well as the perisynaptic astroglial protein ezrin⁷⁰. They are also immunopositive for VGlut1 and Homer1⁷⁰.”

Page 6, lines 10-11: what is the signaling mechanism that triggers the HC function of Cx43 to mediate glutamine release?

To address this question, we designed a series of EtBr uptake experiments using specific pharmacological blockers and genetic knockout mice, and are now able to provide further understanding of the mechanism involved in triggering Cx43 HC opening upon synaptic activity. In brief, we found that upon 10Hz stimulation, an elevation of synaptic glutamate acting on neuronal ionotropic glutamate receptors followed by an increase in extracellular K⁺, known to be primarily released by postsynaptic elements after glutamatergic activity (Poolos et al, 1987), and acting on astroglial Kir4.1 channels, trigger Cx43 HC opening. We have now described and illustrated these experiments and new findings in details in our revised manuscript.

See Pages 8-9, line 176-201, figure 5 in *Results* section:

“To elucidate the signaling mechanism involved in the activity-dependent opening of Cx43 HC mediating glutamine release, we focused on the most prominent candidates which are elevated during excitatory synaptic activity, namely glutamate³⁰ and potassium^{31,32}. We found that the activity-dependent opening of astroglial Cx43 HC, induced by Schaffer collateral stimulation and assessed by EtBr uptake, was mediated by activation of ionotropic glutamate receptors, abundantly expressed in

hippocampal neurons, as it was fully inhibited by antagonists of AMPARs (2,3-Dihydroxy-6-nitro-7-sulfamoyl-benzo[f]chinoxalin-2,3-dion, NBQX, 10 μ M) and NMDARs (3-((R)-2-carboxypiperazin-4-yl)-propyl-1-phosphonic acid, CPP, 10 μ M), but not of metabotropic glutamate receptors (LY341195, 20 μ M, Fig. 5d-e). We also found that potassium channels played a critical role in the activity-dependent opening of Cx43 HC, since the potassium channel blocker BaCl₂ (200 μ M) abolished the enhanced EtBr uptake upon stimulation. The potassium effect was specifically mediated by astroglial Kir4.1 channels, as evidenced by the lack of activity-dependent EtBr uptake in astrocytes deficient for Kir4.1. These findings suggest that glutamate and potassium both contribute to the activity-dependent Cx43 HC opening via activation of neuronal ionotropic glutamate receptors and astroglial Kir4.1 channels, respectively. To further and directly assess the involvement of glutamate and potassium in the activation of Cx43 HC, we incubated hippocampal slices with either potassium (2mM) or glutamate (1 μ M) during EtBr uptake, and found that both similarly activated Cx43 HC in a Gap26-sensitive manner (Fig. 5f-g). To determine the relationship between these pathways, we repeated these experiments in hippocampal slices from mice deficient for astroglial Kir4.1 channels, and found that neither potassium nor glutamate, exogenously applied alone, could increase astroglial EtBr uptake in these mice. These data indicate that both pathways are linked, with the glutamate effect being mediated by potassium activation of astroglial Kir4.1 channels. Altogether, these data reveal that astroglial Cx43 is located near synapses, and suggest that its HC function is enhanced by glutamatergic synaptic activity controlling levels of extracellular potassium activating astroglial Kir4.1 channels.”

See in Discussion section:

Pages 17-18, lines 389-397:

“Our observation that RhGln spread from astrocytes to synapses is also inhibited during basal condition (Fig. 6b) suggests that there is a basal glutamine transfer from the astrocytes to the synapses. Based on our identification of the involvement of glutamatergic synaptic activity triggering Kir4.1-dependent opening of Cx43 HC upon synaptic stimulation, we postulate that an activity-dependent elevation of synaptic glutamate first act on ionotropic glutamate receptors from postsynaptic sites. The resulting release of potassium from postsynaptic elements following glutamatergic activity, known to represent the main source (80%) of neuronal potassium release⁴⁶, then in turn activates Kir4.1 channels at perisynaptic astroglial processes resulting in Cx43 HC opening.”

Page 19, lines 416-418:

“Our findings showing an involvement of glutamate in opening Cx43 HCs could explain a previous study showing that an increased glutamate efflux in the hippocampus is associated with the recognition of novel but not familiar objects⁵⁰.”

Page 8, lines 21-22: addition of glutamine at 4 mM rescues genetically- or pharmacologically-induced impairment of Cx43 function. It would be interesting to know the kinetics of the glutamine transporter in neurons and how 4 mM relates to it.

We have now added details regarding the kinetics of glutamine transporter in neurons.

See Pages 16-17, lines 365-371 in *Discussion* section:

“It has been previously reported that glutamine uptake by neuronal glutamine transporters in transfected fibroblasts is saturable and highly sensitive to glutamine with a Michaelis constant (Km) of 489±88 μM at pH 7.4⁴¹. The authors showed that transporter saturation is reached in the presence of 1mM glutamine. Based on this, and considering slower diffusion in brain tissues compared to cultured cells, the addition of 4mM of glutamine exogenously on acute hippocampal slices is likely to saturate neuronal glutamine transporters, assuring sufficient rescue.”

Page 9, line 2-3: In the introduction the authors indicate that without a glutamine recycling process glutamatergic transmission would be exhausted within one minute of basal synaptic activity. The fluorescent probe used has been chemically designed to prevent the conversion of glutamine to glutamate. The question then is how much the probe interferes with normal neuronal glutamate replenishment as this process cannot be provided by the stable fluorescent probe. Some quantitative data should be provided to address this question.

In order to test whether the RhGln probe interferes with normal neuronal glutamate replenishment, we have monitored synaptic transmission in CA1 hippocampal slices and found that it remains stable over the course of 20 mins of RhGln dye loading into a single astrocyte. This indicates that RhGln does not interfere with the neuronal glutamate replenishment from endogenous glutamine. These new measurements are now described and illustrated in the revised manuscript.

See Page 6, lines 128-133, supplementary figure 4 in *Results* section:

“We also verified that CA1 synaptic transmission remained stable over the course of dye loading into a single astrocyte by monitoring fEPSP slope in response to repetitive synaptic stimulation (10Hz) (Supplementary Fig. 4). This suggests that RhGln can be used to monitor the mobilization of glutamine without affecting neuronal glutamate replenishment from endogenous glutamine, important for synaptic activity.”

Page 14, lines 15-16: It would be important to have a quantitative assessment of what % of total Cx43 are present on astrocytic processes around synaptic profiles.

To quantitatively assess the distribution of Cx43 on astrocytic processes around synaptic profiles, we have now taken two separate approaches using either immunofluorescence or electron microscopy. Both methods independently revealed that 38% (immunofluorescence co-localization) and 40% (Cx43 gold grains) of total Cx43 is present on astrocytic processes around synaptic profiles. This quantification has been included and illustrated in the revised manuscript and in figure 4.

See in *Results* section

Page 7, lines 155-158, figure 4:

“To assess Cx43 localization, we used immunolabeling and found that Cx43 protein is expressed throughout hippocampal astrocytes including fine processes away from the cell soma and close to VGlut1-positive glutamatergic presynapses (Fig. 4a). We found that 38.2 of total Cx43 co-localizes with VGlut1 puncta (Fig. 4b).”

Page 8, lines 164-169:

“We found that 40.6% of total immunogold particles labeling Cx43 were present on astrocytic processes in close proximity to synaptic complexes (<300nm away), which is similar to what was observed using immunolabeling (Fig. 4b). Distance analysis further revealed that most of these Cx43 positive gold grains were at a distance of 200-300nm to the synaptic cleft, with a minimum distance of 72.2nm and an average distance of 265.8 ± 115.5 nm to the nearest active zone (Fig. 4e-f).”

See in *Methods* section:

Page 32, lines 728-731:

“For the determination of the % of perisynaptic Cx43, total Cx43 was first determined as the co-localized area between GFAP-eGFP and Cx43. Then, the perisynaptic population of Cx43 was calculated based on the area which is further co-localized with VGlut1 and then normalized to total Cx43.”

Page 31, lines 694-697:

“All Cx43 positive gold grains identified on the images were counted using ImageJ software (National Institutes of Health, USA). The astrocytic profiles that are near or in close proximity can be at the shortest distance membrane to membrane (less than 20nm) with the synaptic profiles composed of the synaptic bouton and the postsynaptic spine.”

See Page 17, lines 385-387, in *Discussion* section:

“Using immunohistochemistry, biochemistry and electron microscopy, we first show that Cx43 is present near hippocampal synapses. We have verified that ~40% of astroglial Cx43 is found near synapses.”

Page 16, lines 17-18: It would be useful to have some examples of diseases in which Cx43 function is altered.

Some examples of diseases in which Cx43 function is altered are now included.

See Page 20, lines 444-446, in *Discussion* section:

“While astroglial Cx43 dysfunction has been implicated in neurodegenerative disorders like Alzheimer’s disease, glioma, and ischemia⁴³, our data also provide an astrocyte basis for cognitive disorders.”

Point-by-point response to Reviewer #2

This is an elegant and thrilling study based on the development and smart application of a novel fluorescent-labeled glutamine analog. The authors dubbed the rhodamine-tagged glutamine molecule, RhGln, and convincingly demonstrated the usability of this tool. Particularly, RhGln was shown to enable the high-resolution visualization of its transfer from astrocytes to presynaptic structures in vitro under basal conditions and upon synaptic stimulation. By several additional smart and well-designed experiments, the authors eventually unveiled the molecular mechanism and physiological role of this directed glutamine transport.

We thank the reviewer for acknowledging the interest, novelty and quality of our work.

I have some minor comments/suggestions to be considered by the authors:

1.) The authors might also show if the diffusion of RhGln toward neighboring cells is RhGln specific. Does the unconjugated rhodamine dye also diffuse in the same manner (extent and kinetics) in this experimental setting?

To address this point, we have now shown that the extent of diffusion in the astroglial network of the unconjugated rhodamine dye (Rh) over 20 mins of dialysis via a patch pipette in a single astrocyte is similar to the one of RhGln in control conditions. However, upon stimulation, while RhGln diffusion in the astroglial network became significantly more restricted, the diffusion of Rh is rather enhanced. This observation of Rh diffusion is expected based on our previous studies showing activity-dependent trafficking of inert dyes in astroglial network (Rouach et al, 2000; Rouach et al 2008; Roux et al 2011). Altogether, these data indicate that the activity-dependent effect on astroglial diffusion observed is specific to RhGln, whereas the basal level of diffusion is not. This new quantification is shown and discussed in the revised manuscript.

See Pages 5-6, lines 111-119, figure 2 in *Results* section:

“To test whether this redistribution is specific to RhGln, we dialyzed astrocytes with the rhodamine (Rh) dye alone (0.8mM) and observed in basal conditions that Rh also diffused into neighboring cells to a similar extent as RhGln over 20 mins (Fig. 2a and 2e). However, upon enhanced activity (10Hz, 30s), Rh diffusion was significantly enhanced, unlike the restriction observed using RhGln (Fig. 2a and 2f). This is reminiscent of previous observations of an activity-dependent trafficking of inert dyes in astroglial network²²⁻²⁴. Together, these data indicate that the activity-dependent redistribution of glutamine away from the astroglial network is specific for RhGln, as it is not observed with the unconjugated Rh dye.”

See Page 14, lines 305-311 in *Discussion* section:

“Of note, the activity-dependent restriction of intracellular dye transfer into neighboring astrocytes was unique to RhGln, and not observed using the Rh dye alone. In addition, the specific RhGln spatial pattern displayed upon activity, consisting of punctate labeling of presynaptic compartments, was also

not observed with the unconjugated Rh dye alone, and was inhibited upon blockade of neuronal glutamine transport. This indicates that upon activity, RhGln enters synaptic compartments via neuronal glutamine transport, in contrast to the inert control dye, and thus retains the specificity of glutamine cellular trafficking.”

2.) The authors might better elaborate and discuss how the conjugation of the relatively large rhodamine molecule impacts the transport and metabolism of the amino acid. What might be important differences in the biochemistry between Gln and RhGln? What needs to be considered when interpreting respective results based on RhGln? What are the real limitations of using RhGln?

Both Gln and RhGln contain the glutamine moiety. RhGln is functionalized at the amide function of glutamine by a rhodamine derivative. The chemical link between rhodamine and glutamine group is an alkyl chain. Therefore, as shown by density functional theory (DFT) calculations using the m062x method and the 6-31G+(d) base with IEFPCM(water) for the solvent, in the case of RhGln, the Rhodamine fluorophore is spaced at an optimal distance from glutamine leaving the glutamine function intact for its involvement in the glutamine-glutamate cycle. It is also important to note that the rhodamine moiety is relatively flat and positively charged. Although these compounds have different solvent accessible volumes (169.56 Å³ for Gln and 737.64 Å³ for RhGln), the choice of rhodamine as a fluorophore was governed by its interesting photophysical properties (absorption wavelength > 550 nm, high fluorescence quantum yield, possibility of high resolution microscopy).

Here, we show that the activity-dependent spread of RhGln into synaptic structures is specific to its glutamine properties, since it is not observed when using Rh alone. Furthermore, RhGln, like endogenous glutamine, can be transferred to neurons via glutamine transporters, as its transfer from astrocytes to neurons is inhibited by the neuronal glutamine transporter blocker MeAIB. This makes RhGln a useful tool to study immediate mobilization of glutamine as its unmetabolized form. It is however not intended to be used to trace further glutamine metabolism or to interfere with excitatory synaptic transmission. The impact of the conjugation of a rhodamine molecule to glutamine and the limitations of using RhGln are now discussed in the revised manuscript.

See Page 4, lines 82-84 in *Results* section:

“The rhodamine moiety is flat and positively charged. According to the density functional theory (DFT) calculations, it is spaced at an optimal distance from glutamine leaving the glutamine function intact for its biological functions.”

See Page 26, lines 576-582 in *Methods* section:

“Density functional theory (DFT) calculations were performed using the m062x method and the 6-31G+(d) base with IEFPCM(water) for the solvent. RhGln is functionalized at the amide function of glutamine by a rhodamine derivative. The chemical link between rhodamine and glutamine group is an alkyl chain. Although these compounds have different solvent accessible volumes (169.56Å³ for Gln and 737.64Å³ for RhGln), the choice of rhodamine as a fluorophore was governed by its interesting

photophysical properties (absorption wavelength >550 nm, high fluorescence quantum yield, possibility of high resolution microscopy)."

See Page 13, lines 289-297 in *Discussion* section:

"We have indeed engineered a fluorescent tag on the amide side chain at an optimal distance from glutamine. As a result, this conjugation prevents conversion of glutamine to glutamate, while leaving the glutamine function intact for its biological functions. As glutamine is readily metabolized intracellularly, this aspect of our probe is essential to ensure unambiguous visualization of glutamine mobilization. It also allows us to observe the accumulation of glutamine at its site of storage or conversion. This makes RhGln a useful tool to study immediate mobilization of glutamine as its unmetabolized form. It is however not intended to be used to trace further glutamine metabolism or to interfere with excitatory synaptic transmission."

3.) The authors might additionally discuss the potential clinical/medical implications of their findings that show the restoring effects of supplementing synapsis with exogenous glutamine.

Some potential clinical/medical implications of using exogenous glutamine supplements is now discussed in the revised manuscript.

See Page 20, lines 447-456 in *Discussion* section:

"Furthermore, it has been shown that a disrupted excitatory/inhibitory balance is associated with autism spectrum disorders (ASD)⁵² and that the disruption of the glutamate-glutamine cycle may trigger the disease onset⁵³. Interestingly, astrocyte dysfunction have also been linked to ASD⁵⁴ where plasma glutamine level is found to be reduced in patients⁵⁵. Cognitive tasks like memory encoding⁵⁶ and consolidation⁵⁷ also depend on glutamine and major components of the glutamate-glutamine cycle. Its dysregulation during development has also been shown to have long-lasting impact on spatial memory in adult mice⁵⁸. Our findings showing that exogenous glutamine can rescue defects in synaptic transmission caused by the lack of astroglial supply of glutamine would support the use of glutamine as a supplement to control diseases like ASD, as well as to treat or prevent memory impairment."

Point-by-point response to Reviewer #3

This paper by Cheung et al. studies glutamine dynamics on astrocyte networks and glutamine transfer from astrocytes to neurons. For this matter, the authors have developed a novel tool that allows them to visualize glutamine and trace it (a fluorescent form of glutamine that they call RhGln). They perform imaging studies combined with electrophysiology and behavior and found that glutamine transfer from astrocytes to neurons is necessary for a correct vesicle replenishment to maintain the presynaptic glutamate release. I believe this paper represents an advance in the field, and sheds light in the glutamine-glutamate cycle. In general, I consider that the experimental approach is appropriate and that the main conclusions are in accordance with the data presented, however, I have some comments and suggestions that I hope will help to strengthen the manuscript:

We thank the reviewer for acknowledging the novelty of our work and the appropriateness of our experimental approach.

The authors claim that RhGln-labeling is not due to extracellular uptake as bulk loading of slices with RhGln did not label astrocytes. However, the chosen image in Suppl figure 2 to illustrate this is not very convincing. The GFAP-GFP labeling is very sparse and it's difficult to know if the red labeled cells in the stratum radiatum are astrocytes or interneurons. In this point, I think it is also important to point out that neurons do uptake RhGln from the extracellular space, as we can see in the images.

We have repeated the experiment using Aldh1l1-eGFP transgenic mice, which are known to have fluorescence labeling in the majority of astrocytes, compared to GFAP-eGFP mice, where only ~50% are eGFP-labeled in the hippocampus (Yu et al 2020). Using both transgenic mouse lines, we found that none of the 115 GFAP- and 135 Aldh1l1-eGFP astrocytes were loaded with RhGln. We also took advantage of the sparse labeling of astrocytes with eGFP in GFAP-eGFP mice and verified that no RhGln loading was observed at the level of astroglial fine processes, and instead only localized to neurons. These data are now incorporated in our revised manuscript.

See Page 5, lines 98-104, supplementary figure 2 in *Results* section:

“Using Aldh1l1-eGFP transgenic mice, in which the majority of astrocytes is labeled by GFP²¹, we found that while CA1 pyramidal neurons take up RhGln, none of the 135 labeled astrocytes in the hippocampal CA1 region were loaded with RhGln (Supplementary Fig. 2a). In addition to this, no RhGln-labeling at the level of astroglial fine processes was found when loading was performed in acute slices from GFAP-eGFP mice, where these fine processes are visible by GFP-labeling (Supplementary Fig. 2b-d).”

In experiments in Fig2d-e the authors found that synaptic stimulation decreases the glutamine intercellular transfer. But it is possible that this effect is not specific for RhGln, in fact, is it possible that this effect is due to the closing of the hemichannels during the stimulation? A control experiment loading rhodamine would be necessary.

To address this point, we have repeated the same experiment, but instead of loading the RhGln dye into astrocytes for 20 mins, we loaded the unconjugated rhodamine dye (Rh). By doing so, we found that the synaptic stimulation did not decrease Rh intercellular transfer, as measured by the number of surrounding cells labeled with Rh, in contrast to what we found for the RhGln dye. We actually observed a significant increase in network size with the unconjugated Rh dye. Together with the absence of stimulation-dependent spread of Rh into neighboring synaptic structures from loaded astrocytes, this suggests that the activity-dependent mobilization of RhGln is specific to glutamine, but not to the closing of gap junction channels. This is in line with our previous work showing that neuronal activity opens gap junction channels (Rouach et al, 2000; Rouach et al 2008; Roux et al 2011). This new quantification is shown and discussed in the manuscript and in Fig. 1. Note that we assume the reviewer was referring to the closing of “gap junction” channels rather than “hemichannels” when discussing intercellular transfer.

See Pages 5-6, lines 111-119, figure 2 in *Results* section:

“To test whether this redistribution is specific to RhGln, we dialyzed astrocytes with the rhodamine (Rh) dye alone (0.8mM) and observed in basal conditions that Rh also diffused into neighboring cells to a similar extent as RhGln over 20min (Fig. 2a and 2e). However, upon enhanced activity (10Hz, 30s), Rh diffusion was significantly enhanced, unlike the restriction observed using RhGln (Fig. 2a and 2f). This is reminiscent to previous observations of an activity-dependent trafficking of inert dyes in astroglial network²²⁻²⁴. Together, these data indicate that the activity-dependent redistribution of glutamine away from the astroglial network is specific for RhGln, as it is not observed with the unconjugated Rh dye.”

See Page 14, lines 305-311 in *Discussion* section:

“Of note, the activity-dependent restriction of intracellular dye transfer into neighboring astrocytes was unique to RhGln, and not observed using the Rh dye alone. In addition, the specific RhGln spatial pattern displayed upon activity, consisting of punctate labeling of presynaptic compartments, was also not observed with the unconjugated Rh dye alone, and was inhibited upon blockade of neuronal glutamine transport. This indicates that upon activity, RhGln enters synaptic compartments via neuronal glutamine transport, in contrast to the inert control dye, and thus retains the specificity of glutamine cellular trafficking.”

For Fig 4f the manuscript would benefit if the authors provide an explanation for the use of ethidium bromide. For this experiment the authors claim that “this uptake was mediated by Cx43 HCs, as it was abolished by the Gap26 peptide” but it is not abolished, it is reduced compared to control conditions. In my opinion the fact that Gap26 peptide reduces uptake below control conditions is important as it indicates that there is a basal glutamine transfer from the astrocytes to the synapse.

We have now rephrased our text to clarify that it is in fact the enhanced uptake by stimulation that was inhibited by the Gap26 peptide. Indeed, Gap26 alone reduced uptake compared to control conditions, which supports our previous findings that Cx43 HCs are actually open in basal condition

(Chever et al 2014 J Neurosci). Together with our observation that RhGln spread from astrocytes to synapses is also inhibited during basal condition (Fig. 6b), this suggests that there is a basal glutamine transfer from the astrocytes to synapses. This has been incorporated into and discussed in our revised manuscript.

See Page 8, lines 170-176 in *Results* section:

“To test whether physiological synaptic stimulation alters Cx43 HC activity, we performed ethidium bromide (EtBr) uptake assays in acute slices, and observed that synaptic stimulation (10Hz, 30s) markedly increased EtBr uptake (Fig. 5a-c). This enhanced uptake by stimulation was mediated by Cx43 HCs, as it was abolished by the Gap26 peptide, a specific blocker of Cx43 HCs, but not by a scrambled peptide (Src, Fig. 5a-c). In addition, we observed that the Gap26 peptide alone decreased EtBr uptake significantly below control condition, which is in line with our previous findings showing that Cx43 HCs are open in basal condition²⁷”

See Pages 17-18, lines 388-391 in *Discussion* section:

“Here, we also showed that Cx43 HC are open during basal condition as previously reported²⁷. Our observation that RhGln spread from astrocytes to synapses is also inhibited during basal condition (Fig. 6b) suggests that there is a basal glutamine transfer from the astrocytes to the synapses.”

They have found that glutamine transfer is necessary for replenishment of the presynaptic pool of glutamate, and they have done this under a specific synaptic stimulation, but they have not tested whether there is a change in basal synaptic transmission. Therefore, sentences like “Importantly, the same exogenous glutamine treatment in +/- slices had no effect on excitatory transmission (Fig. 7h)” or “we identified astroglial Cx43 HCs as the key mediators of this process, essential for sustaining physiological excitatory synaptic transmission as well as recognition memory” are somehow misleading. One possibility is to perform experiments where they analyze amplitude and frequency of spontaneous or miniature EPSCs, otherwise I feel the authors should down tone the manuscript regarding the role of glutamine on excitatory transmission.

To address this, we have now measured basal synaptic transmission in response to a single stimulus of low intensity (10-20 μ A, 0.1ms), classically used to assess basal transmission with input-output curves (Herron and Malenka, 1994, Fremeau et al 2004, Pannasch et al 2014), and compared the same experimental conditions used in the repetitive stimulation experiments. We found a similar dependency to Cx43 HC activity, which was rescued by exogenous glutamine. We thus conclude that glutamine supply via Cx43 HC is important for basal evoked synaptic transmission at physiological range. This is now illustrated and discussed in our revised manuscript. We now also emphasize in various places of our manuscript that our data describes the role of glutamine in synaptic transmission in basal condition and evoked by repetitive stimulation in the physiological range.

See Page 11, lines 239-246, supplementary figure 5 in *Results* section:

“Furthermore, either Cx43 genetic disruption in astrocytes (-/-) or acute pharmacological inhibition of Cx43 HC by Gap26 in +/+ slices also led to an impairment in basal synaptic transmission, evoked by a single Schaffer collateral stimulation of low intensity (10-20µA, 0.1ms), as revealed by fEPSP slope/FV amplitude (Supplementary Fig. 5). Similar to the results obtained with repetitive stimulation, exogenous glutamine (4mM for 1-4h) also fully rescued the basal synaptic transmission in +/+ slices treated with Gap26 or in -/- slices (Supplementary Fig. 5). Altogether, these findings indicate that Cx43 HCs contribute to physiological excitatory synaptic transmission by fueling synapses with glutamine.”

See Page 16, lines 352-354, Discussion section:

“This dependency on Cx43HC and glutamine was also observed for basal synaptic transmission in response to a single Schaffer collateral stimulation, consolidating the role of glutamine in physiological synaptic transmission.”

References

- Chever, O., Lee, C. Y. & Rouach, N. Astroglial connexin43 hemichannels tune basal excitatory synaptic transmission. *J Neurosci* **34**, 11228-11232, doi:10.1523/jneurosci.0015-14.2014 (2014).
- Fremeau, R. T., Jr. *et al.* Vesicular glutamate transporters 1 and 2 target to functionally distinct synaptic release sites. *Science* **304**, 1815-1819, doi:10.1126/science.1097468 (2004).
- Herron, C. E. & Malenka, R. C. Activity-dependent enhancement of synaptic transmission in hippocampal slices treated with the phosphatase inhibitor calyculin A. *J Neurosci* **14**, 6013-6020 (1994).
- Pannasch, U. *et al.* Connexin 30 sets synaptic strength by controlling astroglial synapse invasion. *Nat Neurosci* **17**, 549-558, doi:10.1038/nn.3662 (2014).
- Poolos, N. P., Mauk, M. D. & Kocsis, J. D. Activity-evoked increases in extracellular potassium modulate presynaptic excitability in the CA1 region of the hippocampus. *J Neurophysiol* **58**, 404-416, doi:10.1152/jn.1987.58.2.404 (1987).
- Rouach, N., Glowinski, J. & Giaume, C. Activity-dependent neuronal control of gap-junctional communication in astrocytes. *J Cell Biol* **149**, 1513-1526, doi:10.1083/jcb.149.7.1513 (2000).
- Rouach, N., Koulakoff, A., Abudara, V., Willecke, K. & Giaume, C. Astroglial metabolic networks sustain hippocampal synaptic transmission. *Science* **322**, 1551-1555, doi:10.1126/science.1164022 (2008).
- Roux, L. *et al.* Astroglial Connexin 43 Hemichannels Modulate Olfactory Bulb Slow Oscillations. *J Neurosci* **35**, 15339-15352, doi:10.1523/JNEUROSCI.0861-15.2015 (2015).
- Yu, X., Nagai, J. & Khakh, B. S. Improved tools to study astrocytes. *Nat Rev Neurosci* **21**, 121-138, doi:10.1038/s41583-020-0264-8 (2020).

REVIEWERS' COMMENTS

Reviewer #1 (Remarks to the Author):

The authors have done a commendable effort in addressing the questions raised by all three reviewers, including performing a large number of novel experiments. Specifically they have addressed the questions that I had raised. The actual molecular mechanism that links synaptic activity to opening of is still not demonstrated. All the authors is that synaptic activity resulting in increased glutamate release and activation of glutamate post-synaptic receptors results in increase extracellular K⁺ that is cleared by K⁺ uptake through astrocytic Kir4.1. This sequence is hardly novel but seems to be necessary to trigger the release of glutamine through Cx43 HC. These observations are only correlative and do not provide the actual molecular mechanism of Cx43 HC activation. Nevertheless there is now sufficient novelty for this article to deserve publication

Reviewer #2 (Remarks to the Author):

All points are adequately addressed. Congratulation, this is a beautiful manuscript.

Reviewer #3 (Remarks to the Author):

The authors have satisfactorily addressed all my comments. Particularly, I think the addition of additional data regarding basal synaptic transmission strengthen the manuscript and their conclusions. I don't have further suggestions and, in my opinion, the manuscript is now acceptable for publication.